behaviour, cognition, neuroscience

social network, social brain, brokerage, fMRI, diversity, attitude

**Author for correspondence:**
Yoosik Youm
e-mail: yoosik@yonsei.ac.kr

# Neural and social correlates of attitudinal brokerage: using the complete social networks of two entire villages

Yoosik Youm[1], Junsol Kim[1], Seyul Kwak[2] and Jeanyung Chey[3]

[1]Department of Sociology, Yonsei University, Seoul, Republic of Korea
[2]Seoul National University Seoul Metropolitan Government Boramae Medical Center, Seoul, Republic of Korea
[3]Department of Psychology, Seoul National University, Seoul, Republic of Korea

YY, 0000-0003-3822-5777

To avoid polarization and maintain small-worldness in society, people who act as attitudinal brokers are critical. These people maintain social ties with people who have dissimilar and even incompatible attitudes. Based on resting-state functional magnetic resonance imaging ($n = 139$) and the complete social networks from two Korean villages ($n = 1508$), we investigated the individual-level neural capacity and social-level structural opportunity for attitudinal brokerage regarding gender role attitudes. First, using a connectome-based predictive model, we successfully identified the brain functional connectivity that predicts attitudinal diversity of respondents' social network members. Brain regions that contributed most to the prediction included mentalizing regions known to be recruited in reading and understanding others' belief states. This result was corroborated by leave-one-out cross-validation, fivefold cross-validation and external validation where the brain connectivity identified in one village was used to predict the attitudinal diversity in another independent village. Second, the association between functional connectivity and attitudinal diversity of social network members was contingent on a specific position in a social network, namely, the structural brokerage position where people have ties with two people who are not otherwise connected.

## 1. Introduction

Many real-world networks are small-world networks in which (i) nodes are locally clustered and (ii) inter-cluster ties enable every node in a network to be linked within short chains of nodes [1–3]. This 'small-worldness' characteristic has been observed in numerous kinds of networks: social, technological, physical and biological networks [3], including acquaintance networks between people in the United States of America [4], scientific collaboration [5], the internet [6], the power grid [1], airline traffic [7], the structure and conformation space of polymers [8], metabolic pathways [9], and brain networks [10].

Brokerage in social network analysis refers to the occupation of inter-cluster ties in a social network [11], which is necessary for a small-world network to emerge. In this study, we define 'attitudinal brokerage' as the occupation of inter-cluster ties between clusters of people with dissimilar attitudes. In modern society, people with similar attitudes tend to cluster in many ways: urban residential neighbourhoods are segregated [12], people with similar attitudes are clustered in Twitter and political blog networks [13,14], and partisan television news networks are becoming popular [15]. Attitudinal brokerage across such clusters can bridge diverse attitudes within short chains of acquaintances, reducing political polarization and promoting social integration [16].

Considering the essential role of attitudinal brokerage, we examined the neural and social characteristics that might account for attitudinal brokerage, measured as

the 'attitudinal diversity of one's social network members.' With regard to neural characteristics, we targeted mentalizing brain regions related to humans' ability to infer other people's attitudes, belief states and intentions by incorporating various social cues [17–21]; these regions include the dorsomedial prefrontal cortex (dmPFC), temporoparietal junction (TPJ) and precuneus. Mentalizing brain regions might contribute to the attitudinal brokerage by resolving two distinct cognitive challenges of coping with social network members' diverse attitudes. First, mentalizing regions may be implicated in processing dissimilarity between 'the self and a friend' (i.e. dyadic challenges) [18,22]. For example, mentalizing regions were activated when liberal people try to extrapolate conservative people's opinions, likes and dislikes [18]. Also, mentalizing regions are known to be activated when one's expectation about other people's attitudes are violated [22]. Second, mentalizing regions may be implicated in processing dissimilarity between 'two friends of oneself' (i.e. triadic challenges). For example, mentalizing regions are implicated in one's ability to connect with friends who are affiliated in different social groups [21], which are likely to have different attitudes.

Many studies that implied the role of mentalizing brain regions in attitudinal brokerage are task-based studies. However, some of the task-free studies also revealed that mentalizing brain regions might be associated with the attitudinal brokerage. For example, mentalizing regions' intrinsic functional connectivity at rest was associated with the size of social networks [23,24]. Also, mentalizing regions' structure (e.g. grey matter volume, white matter integrity) was associated with the size and diversity of social networks [25–28]. These studies are in line with previous literature that brain-behaviour associations are driven in part by stable trait-level variation in intrinsic brain connectivity [29,30]. Therefore, we hypothesized that intrinsic brain connectivity from and to mentalizing regions would be correlated with attitudinal diversity of social network members, an indicator of attitudinal brokerage.

In addition to the neural correlates of attitudinal brokerage, which refers to the individual-level intrinsic capacity to maintain social ties with heterogeneous attitudes, we suggest that social-level structural opportunity for being exposed to people with diverse attitudes is also necessary for attitudinal brokerage. Even if somebody has the neural capacity to bridge people with diverse attitudes, he or she still needs to be surrounded by people with different attitudes to exercise that neural capacity. Thus, we investigated whether the role of brain connectivity is contingent on the social opportunity, i.e. having connections with diverse attitudes. According to social network analysis, there exist two ideal types of positions relevant to such opportunity: structural closure and structural brokerage (electronic supplementary material, figure S1). In a structural closure position, where one's social ties tend to be friends with each other, one is probably constrained by a strong unified social norm and is more likely to share similar attitudes with connected others [11,31–34]. On the contrary, in a structural brokerage position, where one's friends are strangers to each other, people are more likely to be exposed to increasingly diverse attitudes, and their autonomy and power to control and moderate dissimilar and even incompatible attitudes increases [11,35–37]. Therefore, we investigated whether the correlation between brain connectivity of mentalizing regions and attitudinal brokerage is contingent on the social opportunity provided in a structural brokerage position.

Thus, we had two major goals in this study. First, we identified the individual-level brain connectivity of mentalizing regions that could reliably predict the attitudinal diversity of social network members. Second, we examined whether the association between identified brain connectivity and attitudinal diversity of social network members is contingent on the structural brokerage position in social networks.

For our first research goal, resting-state brain functional magnetic resonance imaging (fMRI) data of a sub-population from the two villages were used ($n = 139$). We implemented connectome-based predictive modelling (CPM) to predict attitudinal diversity of social network members from resting-state brain functional connectivity of mentalizing regions [38]. In particular, we used global brain connectivity from mentalizing regions to other whole-brain regions for prediction, considering that high-level processing such as social processing employs a large-scale network of brain regions rather than isolated areas [39,40]. To evaluate the predictive accuracy, we implemented two cross-validation procedures: leave-one-out cross-validation (LOOCV) and fivefold cross-validation. Furthermore, control analyses and a reproducibility check were employed to examine the robustness of predictive accuracy. Then, for the second research goal, we conducted a social network analysis that examined if the association between resting-state brain functional connectivity of mentalizing regions and attitudinal diversity of social network members is contingent on occupying a structural brokerage position in social networks.

## 2. Materials and methods

### (a) Acquisition of global (or complete) social network data

Social network data used in this study were from the Korean Social Life, Health and Aging Project (KSHAP), which is a community-based cohort study on an entire population who were 60 or older and their spouses in two Korean rural villages (townships K and L in Ganghwa-Island, Incheon, South Korea) [41,42]. Because almost all of the older adults residing in the two villages ($n = 1508$; township K $n = 562$, township L $n = 946$, response rate = 85.97%) went through the social network survey based on face-to-face interviews, we were able to capture a relatively accurate picture of every intimate social relationship within the two villages as shown in figure 1. During the social network survey, participants indicated their 'discussion partners', including a spouse, if any, and up to five people with whom they most often discussed important personal concerns over the last 12 months. For each discussion partner, the participant provided a real name, age, gender, address of residence and communication frequency (days per year). We constructed an undirected social network where the tie between two people is assumed if at least one person nominates another person as a discussion partner. It is because our survey data measured very strong social ties: 'spouse and top-five important discussion partners'. Therefore, we assumed that both 'someone who is indicated by me as important discussion partner' and 'someone who indicates myself as an important discussion partner' are important social network members.

### (b) Measurements of gender role attitudes and attitudinal diversity scores

Gender role attitudes for each participant were estimated using two items from the 2002 International Social Survey Program (ISSP) module 'Family and Changing Gender Roles III' [43].

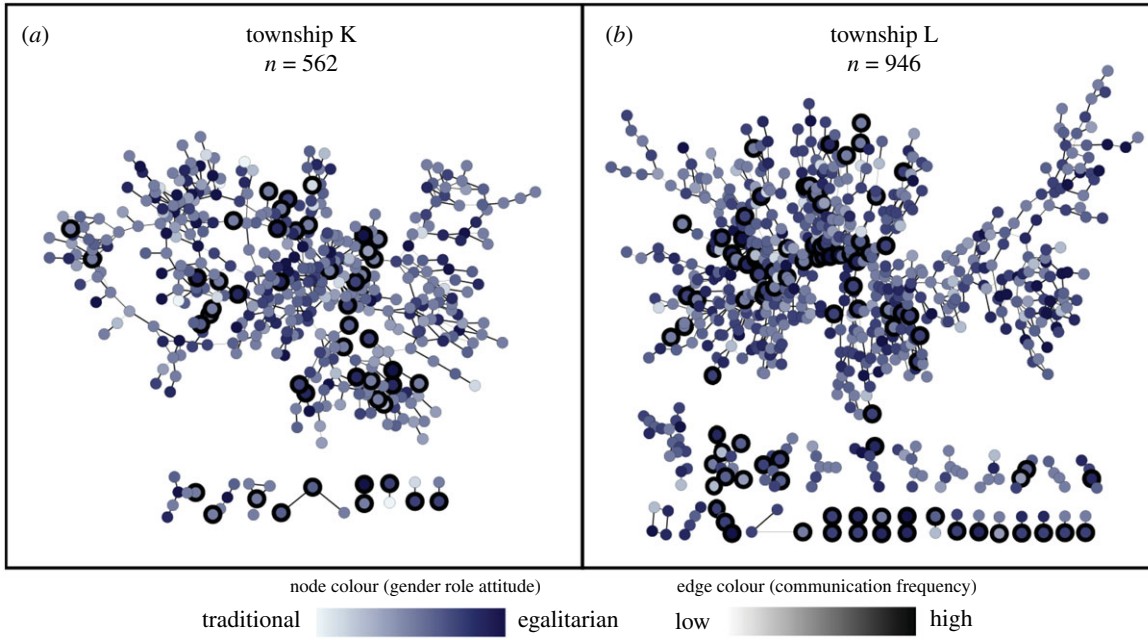

**Figure 1.** Global (or complete) social networks of two entire villages in which study participants who completed fMRI (marked by black bold circles) resided. Each complete social network was constructed using survey data from almost all of the older adults residing in each village (response rate = 85.97%). Each node (circle) represents one person, and each edge (line) represents the relationship between two people. Node colour indicates one's gender role attitudes, and edge colour indicates communication frequency (d yr$^{-1}$) between two people. (Online version in colour.)

Participants were asked whether they agree or disagree with two statements: 'Both the man and woman should contribute to the household income' and 'A man's job is to earn money; a woman's job is to look after the home and family'. Responses ranged from 1 (strong agreement) to 5 (strong disagreement). We reverse-coded the response to the first question such that higher scores indicated more egalitarian gender role attitudes for each question. The gender role attitude score was the average of the recoded responses to the first and second questions. The two items have been the most frequently used gender role attitude measures [44]. The reliability estimate of the two items was also high in the previous studies [45]. However, in our study, the reliability estimate of the two items was not so high (Spearman–Brown coefficient = 0.222). We suspect that the first and the second questions may capture different dimensions of gender egalitarianism in the context of modern Korean rural villages. Nevertheless, we used the composite score because we believe both dimensions are essential components of gender egalitarianism. Communication frequency indicates how many days each social network member and the participant talked to each other (including face-to-face conversation and phone calls) per year.

We calculated attitudinal diversity scores of social network members using two metrics: ambivalence scores and standard deviation. Note that participants were included as a social network member, and communication frequency with him/herself per year was assumed to be 365 [46]. First, we calculated weighted ambivalence scores as follows [47]: $T$ refers to the total communication frequency (sum of communication frequencies) with social network members whose gender role attitudes score was 3 or lower, and thus, whose attitudes could be assumed as traditional. $E$ refers to total communication frequency with egalitarian social network members, whose gender role attitudes score was greater than 3. Note that response categories to gender role attitudes score are 1 = strongly agree, 2 = agree, 3 = neither agree nor disagree, 4 = disagree, 5 = strongly disagree and thus, the midpoint is 3:

$$\text{ambivalence} = \frac{T + E}{2} - |T - E|.$$

Therefore, higher ambivalence score would indicate more diverse attitudes of social network members (i.e. connecting with both

traditional and egalitarian people). Given that the maximum number of friends are 10 (see the electronic supplementary material, table S1), the maximum ambivalence score is 1825 ( = (365*5 + 365*5)/2-|365*5–365*5|) and the minimum score is −1825 ( = (365*10 + 0)/2-|365*10–0|).

Second, to calculate standard deviations, we calculated a weighted standard deviation of gender role attitudes among each participant's social network members as follows. For each social network member, $x$ indicates his or her gender role attitude score, and $d$ indicates communication frequency with him/her:

$$\text{standard deviation} = \sqrt{\frac{\sum d(x - (\sum d \times x / \sum d))^2}{\sum d}}.$$

Therefore, the higher standard deviation would indicate more diverse attitudes of social network members. The biggest difference between the two metrics is that the ambivalence score is based on the bi-variate measure, $T$ or $E$, while the standard deviation is based on continuous value. Before the statistical analyses, we converted both scores to standardized $z$-scores. Participants differed widely with respect to the attitudinal diversity of social network members (electronic supplementary material, figure S2). A sub-population of the original participants was scanned for the study ($n = 139$, mean age = 72.73, 54.68% females). The electronic supplementary material, table S1 presents the descriptive statistics of the participants. Also, as shown in the electronic supplementary material, table S2 and figure S3, traditional people were more likely to connect with people having diverse attitudes. We believe, as shown in the electronic supplementary material, figure S4, the plausible explanation would be that there were more egalitarian people than traditional people in our sample.

## (c) Acquisition of resting-state functional magnetic resonance imaging data and image processing

Resting-state fMRI data ($n = 139$; township K = 47, township L = 92) were acquired from a sub-population of the KSHAP. To select participants for resting-state fMRI, screening tests were conducted as described in the electronic supplementary material,

text S1, Screening tests. After collecting and processing the images as described in the electronic supplementary material, text S2, Acquisition of resting-state fMRI data and image processing, we constructed 139 individual whole-brain connectivity matrices containing 25 651 (=(227 × (227–1))/2) edges, which signify the coherence between every pair of 227 brain regions from Shen's whole-brain parcellation atlas. The study was approved by and performed in accordance with the relevant guidelines and regulations by the Institutional Review Board of Yonsei University (IRB number: YUIRB-2011-012-01; township K survey, brain fMRI and township L survey) and the Institutional Review Board of Seoul National University (IRB number: 1801/001-003; township L brain fMRI), and all participants provided written informed consent for the research procedure.

## (d) Mentalizing brain network

We identified 51 mentalizing-related brain regions among the 227 regions from Shen's whole-brain parcellation atlas. To do so, we used a mentalizing mask, which is the association test map of the term 'mentalizing' from the Neurosynth meta-analytic tool (http://neurosynth.org/analyses/terms/mentalizing/) (see the electronic supplementary material, text S3, Mentalizing brain network). All 51 regions are shown in the electronic supplementary material, figure S5. Given these 51 'mentalizing' brain regions, we identified a mentalizing brain network consisting of 10 251 edges between the 51 'mentalizing' regions and all 227 whole-brain regions, in line with previous studies [40].

## (e) Measurement of social network size and Burt's structural constraint

Social network size was measured as the number of people connected to each participant in the village-wise complete social networks. Second, Burt's structural constraint was calculated as an inverse measure of the occupation of a structural brokerage position given by the following equation [11]: $p_{ij}$ corresponds to the time and resources invested by person $i$ into a social relationship with other person $j$. $p_{ij}$ is the communication frequency signifying how many days the participant $i$ and his or her social network member $j$ communicated for a year divided by the total communication frequency with all of person $i$'s social network members. Multiplying $p_{iq}$ and $p_{qj}$ approximates the indirect constraint imposed on person $i$ by person $j$ via the third actor, $q$, who has relationships with both $i$ and $j$. Thus, in the below equation, the first term measures the structural constraint imposed on person $i$ through his/her dyadic relationship with person $j$: the more time and resources person $i$ spends on the relationship with person $j$, the more constraint is imposed on person $i$. The second term in the equation measures the structural constraint imposed through triadic relationships: the more time and resources person $i$ spends on person $q$, who in turn invests resources on person $j$, the greater the constraint is on person $i$ by person $j$. Therefore, if person $i$ connects many others who are otherwise unconnected (i.e. occupies a structural brokerage position), person $i$'s structural constraint value should be low. It should be noted that a structural brokerage position is conceptually different from a hub node (see the electronic supplementary material, text S4, Structural brokerage and hub node):

$$\text{constraint}_i = \sum \left( p_{ij} + \sum p_{iq} p_{qj} \right)^2 .$$

## (f) Connectome-based predictive modelling

To identify the association between brain connectivity and attitudinal diversity scores, this study applied CPM. CPM is a method based on a machine learning approach to predict individual attributes based on resting-state brain functional connectivity, and the MATLAB code used for CPM is freely available online (https://www.nitrc.org/projects/bioimagesuite/) [38,48]. To test this association, CPM generally implements LOOCV as described in the electronic supplementary material, text S5, Connectome-based predictive modelling (CPM) [38]. For each LOOCV round, $(n-1)$ participants are used as the training sample to estimate the predictive model, and the remaining one is used as the test sample to evaluate the predictive accuracy of the model. LOOCV rounds were repeated such that each participant was used once as the test sample. To estimate predictive accuracy, which represents the significance of the association, Pearson's correlation coefficient ($r$) and mean absolute error (MAE) between the observed attitudinal diversity scores and model-predicted attitudinal diversity scores were calculated. To account for the non-independence of the leave-one-out rounds, permutation tests were conducted. It should be noted that the permutation test also minimizes the potential statistical inference problem caused by the bimodal distribution of the ambivalence score since the test is computed by comparing the obtained test statistic against the 'permutation', rather than the theoretical distribution of the test statistic based on the normality assumption [49]. In addition to the LOOCV, fivefold cross-validation was also employed. To control for confounding effects, we conducted control analyses as described in the electronic supplementary material, text S7, Control analyses.

## 3. Results

### (a) Neural correlates

We successfully identified mentalizing brain connectivity that significantly and positively predicted both types of attitudinal diversity scores (ambivalence score and standard deviation) (see the electronic supplementary material, text S5, Connectome-based predictive modeling (CPM)). As shown in the electronic supplementary material, figure S6, LOOCV revealed that mentalizing brain connectivity positively predicted ambivalence score ($r = 0.2301$, $p = 0.046$; MAE = 0.8305, $p = 0.030$). Further, mentalizing brain connectivity positively predicted standard deviation marginally ($r = 0.2033$, $p = 0.061$; MAE = 0.8190, $p = 0.114$). On the other hand, neither ambivalence scores ($r = 0.1136$, $p = 0.265$; MAE = 0.8971, $p = 0.274$) nor standard deviations ($r = -0.0182$, $p = 0.557$; MAE = 0.9232, $p = 0.712$) were negatively predicted by mentalizing brain connectivity. As shown in the electronic supplementary material, table S3, mentalizing brain connectivity significantly and positively predicted two types of attitudinal diversity scores even after controlling for confounding variables such as socio-demographic factors (age, sex, years of education), social network characteristics (social network size, average communication frequency, structural constraints, betweenness centrality, close centrality, eigenvector centrality), health (Mini-Mental State Exam (MMSE), subjective health), personality (agreeableness, extraversion, neuroticism, openness to experience, conscientiousness), participant's own gender role attitudes, head motion (maximum frame-wise displacement and mean frame-wise displacement) and township of residence. Following previous studies, we controlled for each confounding variable one-by-one while controlling for age and sex by default [48,50].

Consistent with the LOOCV results, fivefold cross-validation showed that both ambivalence scores ($r = 0.2283$, $p = 0.011$; MAE = 0.8368, $p = 0.006$) and standard deviations

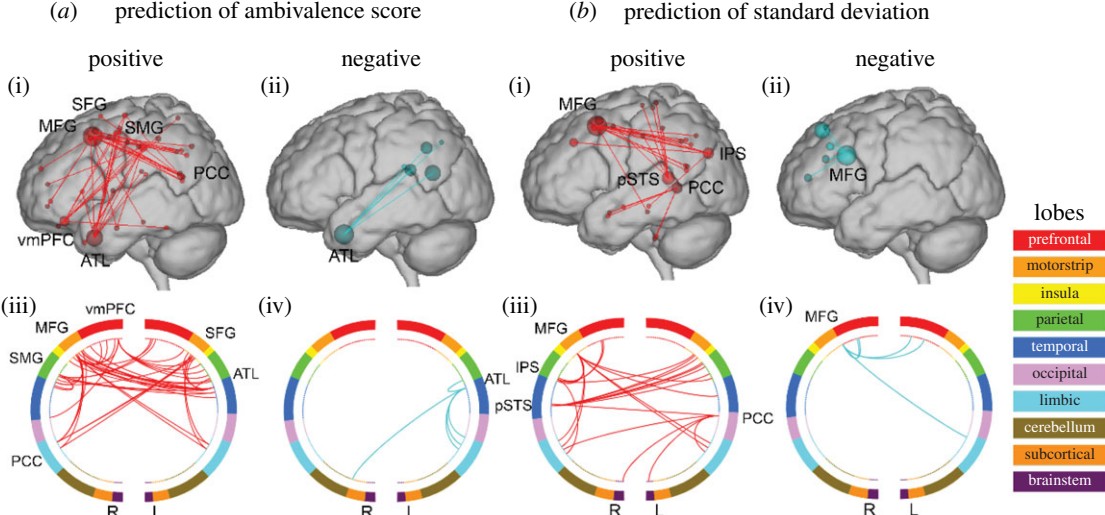

**Figure 2.** Depictions of the predictive mentalizing brain networks Glass brains and circle plots were thresholded to show the highest degree (*k*) nodes in the networks (*k* = 5). Red and blue lines indicate the edges in the mentalizing brain connectivity that positively and negatively predicted ambivalence score or standard deviation. Colours within the circle plots correspond to lobes of the brain (note that the cerebellum was not included). L, left hemisphere; R, right hemisphere; MFG, middle frontal gyrus; pSTS, posterior superior temporal sulcus; IPS, intraparietal sulcus; PCC, posterior cingulate cortex; SFG, superior frontal cortex; vmPFC, ventromedial prefrontal cortex; SMG, supramarginal gyrus; ATL, anterior temporal lobe. (Online version in colour.)

($r$ = 0.1912, $p$ = 0.017; MAE = 0.8110, $p$ = 0.018) were significantly and positively predicted by mentalizing brain connectivity (electronic supplementary material, figure S7). Conversely, neither ambivalence scores ($r$ = 0.0509, $p$ = 0. 265; MAE = 0.9310, $p$ = 0.303) nor standard deviations ($r$ = −0.0413, $p$ = 0.584; MAE = 0.9045, $p$ = 0.556) were negatively predicted by mentalizing brain connectivity.

To complement this hypothesis-driven approach that used mentalizing brain connectivity defined using a meta-analytical approach as a predictor of attitudinal diversity of social network members, we used whole-brain resting-state functional connectivity as a predictor. As a result, we found that the prediction performance of mentalizing brain connectivity was above and beyond whole-brain resting-state functional connectivity (see the electronic supplementary material, text S8, Prediction using whole-brain connectivity). In order to investigate the reproducibility of our results, we conducted a reproducibility check. Given that our participants were from two non-adjacent independent villages, we examined if the predictive performance of mentalizing brain connectivity in one village was successfully replicated in the independent sample of another village. The connectivity of the predictive mentalizing brain network as identified in village L could also successfully predict the standard deviation ($r$ = 0.4042, $p$ = 0.005) in village K. The prediction was not so successful for the case of the ambivalence scores ($r$ = 0.1424, $p$ = 0.340). The result suggests that our results may successfully replicate in an independent sample (see the electronic supplementary material, text S9, Reproducibility check).

## (b) Neuroanatomy of neural correlates

What specific edges in the mentalizing brain network contributed most to the prediction of attitudinal diversity scores? To answer this question, we identified a set of edges for which connectivity values positively predicted attitudinal diversity scores for more than 90% of LOOCV rounds out of the original 10 251 edges. In this study, we called this set of edges the 'predictive mentalizing brain network'.

Because we had two types of attitudinal diversity scores, we also had two kinds of predictive mentalizing brain networks, as visualized in figure 2. First, we identified 78 edges that positively predicted ambivalence score, including the hubs that contributed most to the prediction using the degree (number of edges connected to a region; *K*) of each brain region [29]. Mentalizing regions such as anterior temporal lobe (ATL; *K* = 12), ventromedial prefrontal cortex (vmPFC; *K* = 7), posterior cingulate cortex (PCC; *K* = 5), TPJ (*K* = 4) along with other regions such as middle frontal gyrus (*K* = 14), supramarginal gyrus (*K* = 6), superior frontal gyrus (*K* = 5) and superior parietal lobule (*K* = 4) were hubs (electronic supplementary material, table S4). Second, we also identified 51 edges that positively predicted the standard deviation. Mentalizing regions such as posterior superior temporal sulcus (pSTS; *K* = 6), intraparietal sulcus (*K* = 5), PCC (*K* = 5), dorsolateral prefrontal cortex (dlPFC; *K* = 4), supramarginal gyrus (*K* = 4) along with other regions such as middle frontal gyrus (*K* = 9), and superior frontal gyrus (*K* = 4) were hubs (electronic supplementary material, table S5). Because negative prediction was not statistically significant, we focused on the interpretation of a set of edges for which connectivity values positively predicted attitudinal diversity scores.

## (c) Social correlates

For the second goal of the study—namely, to identify the opportunity structure for attitudinal diversity—we employed social network analysis. Using each participant's mentalizing brain connectivity, we allocated each participant to a high mentalizing brain connectivity group (people with the median or higher connectivity) or a low mentalizing brain connectivity group (people with lower than the median connectivity). Mentalizing brain connectivity was calculated from the total connectivity value of edges in the predictive mentalizing brain network, which is presented in §3b. Using Burt's structural constraint, which is a summary index of structural brokerage and structural closure position

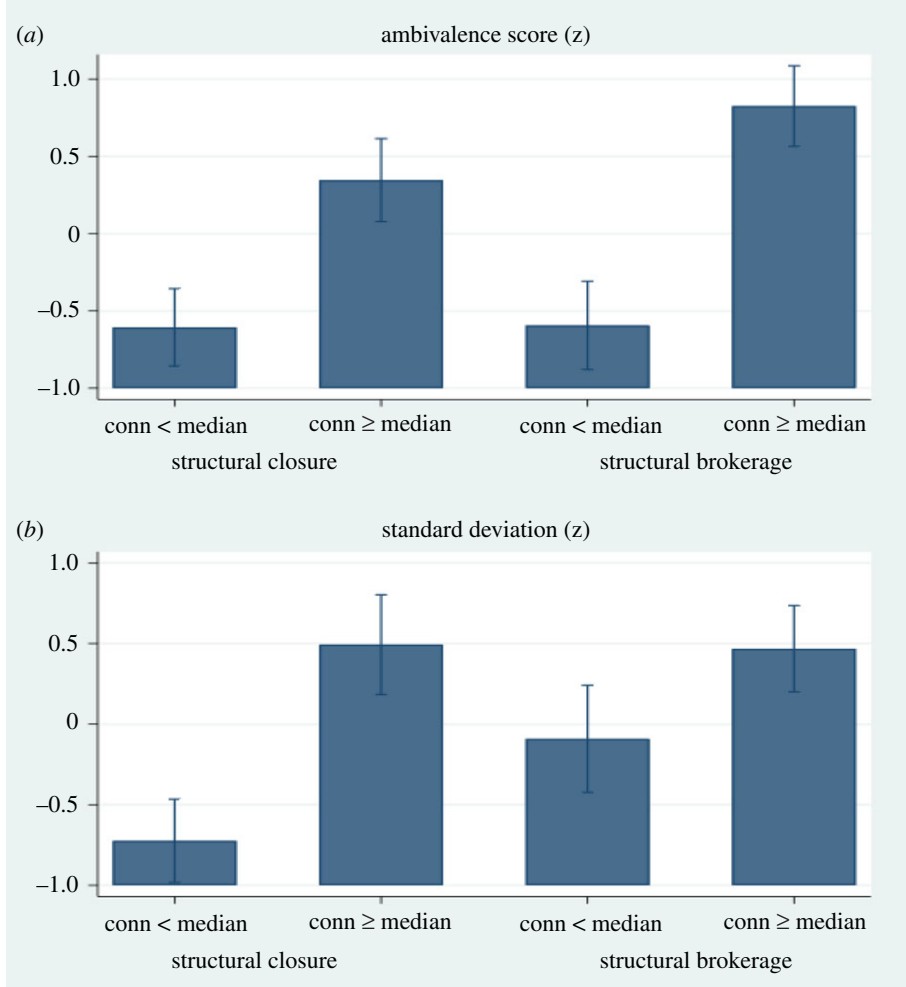

**Figure 3.** Social network analyses. The interaction between high mentalizing brain connectivity and structural brokerage group was marginally significant for ambivalence scores ($p = 0.095$) and significant for the standard deviation ($p = 0.015$). Conn: mentalizing connectivity, structural closure: Burt's structural constraint ≥ median. Structural brokerage: Burt's structural constraint < median. (Online version in colour.)

in a social network, we assigned each participant to a structural brokerage group (structural constraint is lower than the median) or a structural closure group (structural constraint is the median or higher). We used the median rather than the mean to divide participants because structural constraint was highly skewed (electronic supplementary material, figure S8). As shown in figure 3, participants who were simultaneously in the high brain functional connectivity group and the structural brokerage group showed high ambivalence scores and standard deviations. Based on a two-way ANCOVA controlling for variables of no interest (age, sex, social network size, mean communication frequency, MMSE and village), we found that the main effects of structural brokerage group on ambivalence score ($p = 0.749$) and standard deviation ($p = 0.169$) were non-significant. However, the interaction between high mentalizing brain connectivity and structural brokerage group was marginally significant for ambivalence scores ($p = 0.095$) and significant for the standard deviation ($p = 0.015$).

To show that the results were consistent even when using continuous variables, we examined the moderating effects of Burt's structural constraint (continuous) on the association between mentalizing brain connectivity (continuous) and attitudinal diversity of social network members (continuous) (see the electronic supplementary material, text S10, Moderating effects).

## 4. Discussion

The existence of attitudinal brokerage is an earmark of modern culture [51], and it is essential to small-world societies for avoiding polarization. We tried to identify both the individual neural capacity and social structural opportunity necessary for the attitudinal brokerage to emerge based on the complete social networks of two entire villages. One point worth noting is that we were able to measure the attitudinal diversity of social network members based on each respondent's actual self-report on his or her own attitudes, without relying on respondents' guesses about the diversity of social network members' attitudes, which may be inaccurate and biased [52].

### (a) Neural correlates

Some noteworthy mentalizing regions that contributed to the prediction included mentalizing regions such as vmPFC and dlPFC in the prefrontal cortex; anterior temporal lobe and pSTS in the temporal lobe; TPJ, intraparietal sulcus and supramarginal gyrus in the parietal lobe; and PCC in the limbic cortex. Additionally, middle frontal gyrus and superior frontal gyrus along the motor strip as well as superior parietal lobule in the parietal lobe, which have been implicated in resolving cognitive conflicts and

information propagation, were among the key regions that contributed to the prediction [53–59].

We need to resolve two distinct types of cognitive challenge to maintain social ties with people who have various attitudes: first, to deal with dissimilarity between 'the self and a friend of oneself' (dyadic challenges) [60] and second, to bridge dissimilarity between 'two friends of oneself' (triadic challenges) [32]. We believed that the brain regions which contributed to the prediction in our study would interactively cope with each of these cognitive challenges.

First, dissimilar attitudes between 'the self and the other' complicate understanding, evaluating and predicting the behaviours of the other in dyadic relationships [60–62]. Mentalizing regions such as vmPFC, anterior temporal lobe, pSTS, TPJ, intraparietal sulcus, supramarginal gyrus and PCC, which are known to be recruited in understanding others' belief states, would support the cognitive capacity to deal with such challenges [63–65]. In previous studies, these regions have been implicated in understanding heterogeneous others who had unexpected political attitudes or behaved in unanticipated ways [22,66–68]. In addition, when using stereotypical knowledge instead of self-referential processing to understand heterogeneous others, vmPFC is recruited less, implying that low capacity of the vmPFC may inhibit the formation of successful social relationships with diverse others [69].

Second, dissimilar attitudes 'between the others' imposes a cognitive challenge known as 'bridging responsibilities' [32]. Imagine a triadic relationship wherein person A connects with both person B and person C, who have dissimilar attitudes with each other. Merely understanding the attitudes of person B and person C would not be enough for person A to maintain ties with both people. Person A would need to be able to 'switch' between different cognitive frameworks underlying different attitudes [32] so that he or she can naturally communicate with both person B and person C [52]. To resolve conflicts between different attitudes and effectively switch between two attitudes, mentalizing brain regions, such as precuneus, and regions implicated in resolving cognitive conflict when provided with conflicting social cues, such as dlPFC, TPJ, middle frontal gyrus and superior frontal gyrus, might be required [53–55,70]. For example, Chiang et al. [70] found that precuneus, dlPFC, middle frontal gyrus and superior frontal gyrus were recruited when one interacts with two people who are affiliated in different social groups and who have a negative relationship [70]. Moreover, person A might need to transmit high volumes of contradicting information and ideas, which stem from heterogeneous attitudes between person B and person C [32,71]; dlPFC and TPJ may be supportive of such information propagation [56–59]. Our control analyses show that attitudinal diversity score is well-predicted even after participant's own gender role attitude is controlled for. This implies that the identified brain connectivity could be related to a cognitive challenge imposed by dissimilar attitudes 'between the others' (triadic challenge) above and beyond dissimilar attitudes between 'the self and the other' (dyadic challenge).

Many previous studies have confirmed that mentalizing regions help support a large social network [23–26,28,72], casting doubts that our findings may be redundant. However, the predictive mentalizing brain network shown here provides novel insights into understanding social connections with *diverse* others, as opposed to just *many* others. To our knowledge, many of the mentalizing regions that contributed to the prediction in our results have not been associated with social network size (e.g. dlPFC, intraparietal sulcus, supramarginal gyrus). Further, our control analysis showed that the mentalizing brain connectivity predicted attitudinal diversity of social network members even after controlling for the size of social networks. Although brain regions supportive of dyadic relationships (i.e. understanding others) have long been discussed, few studies have discussed the brain regions supportive of triadic relationships, which would be crucially implicated in social relationships with diverse others and further implicated in the emergence of small-world social networks [1]. Our findings could lay the foundation for further studies of how various brain regions interactively support intricate social situations in triadic relationships. The overall results are consistent with prior arguments that mentalizing regions along with various modular brain networks including non-social and domain-general networks interactively relate to one's social interactions [23,26,28,73].

## (b) Social correlates

Social network analyses revealed that the role of mentalizing brain connectivity in connecting diverse others was contingent on a specific social network position: a structural brokerage position where one is more likely to have ties with two people who are not otherwise connected to each other. People who not only had high mentalizing brain connectivity but also occupied structural brokerage positions in a social network were more likely to relate to the most diverse others. Therefore, for mentalizing brain connectivity to be used to the full extent to connect with heterogeneous others, one was required to be on structural brokerage position which provides an opportunity to be exposed to and connect with people who would not be connected otherwise and thus are more likely to have different attitudes. These results offered insights into how neural and social correlates together are implicated in the survival of diverse attitudes in a society. We believe that both individual-level neural capacity (or propensity) and social-level structural opportunity are necessary to avoid polarization of attitudes and values, and achieve and maintain modern small-world societies.

## (c) Limitations

There were a few noteworthy limitations in this study. First, as the research was conducted on older adults who resided in the rural villages in South Korea, the results may not be generalized to the other types of populations. Additional studies examining younger populations and/or urban areas in countries with different cultural and social contexts would be helpful to confirm the generality of our findings. The source code to estimate the model to predict attitudinal diversity of social network members employed in this study, along with the corresponding dataset, is available for the purpose of replication (https://osf.io/azrsy/?view_only=d6dd38ffbd7244e6b0cf9928cdeefe3c).

Second, we measured the social network members using the name generator of 'important discussion members', which were the inner-most core within hierarchical layers of personal social network members [26]. Further investigation of extended layers of social relationships (for example,

free-time partners who spend free time with participants) may be tested in future studies.

Third, we measured attitudinal diversity by using only one attitudinal measure, gender role attitudes, because this was the only attitudinal measure in our dataset. Future studies may use other sets of attitudinal or belief measures to examine whether our results can be generalized (see the full codebook available at https://osf.io/azrsy/?view_only=d6dd38ffbd72 44e6b0cf9928cdeefe3c). Considering that the reliability estimate of our gender role attitude measure was not so high, we examined whether one of the two items in the measure drove our results. However, we found that this was not the case. Both items were statistically significant and the difference between them was not statistically different (see the electronic supplementary material, text S11, Additional analyses for our gender role attitude measure).

Fourth, mentalizing brain regions are also implicated in the default mode network. Therefore, some may want to argue that mentalizing brain connectivity is not related to mentalizing activity but indicates general intrinsic connectivity associated with the brain at rest. However, we still believe that connectivity between mentalizing regions at rest indicates mentalizing processes for two reasons. First, as shown in the electronic supplementary material, tables S4 and S5, mentalizing regions not included in the default mode network, such as pSTS and pre-supplementary motor area, are part of the important hub nodes that contribute most to the prediction of attitudinal diversity. Second, previous task-free studies also showed that intrinsic functional connectivity at rest and structure of mentalizing brain regions are associated with the size and diversity of social networks [23–28]. We would benefit from a future study that elucidates whether our results are also replicated in task-based experiments.

Lastly, owing to the cross-sectional design of this study, causality is unclear. While the connectivity of brain regions could affect the diversity of social network members, it is also possible that the diversity of social network members stimulate and influence the connectivity of brain regions, owing to the cognitive challenge of dealing with dissimilar and sometimes even conflicting attitudes and demands [32]. Future research equipped with longitudinal data would be helpful to shed light on this issue.

Ethics. The study was approved by and performed in accordance with the relevant guidelines and regulations by the Institutional Review Board of Yonsei University (IRB number: YUIRB-2011-012-01) and the Institutional Review Board of Seoul National University (IRB number: 1801/001-003).

Data accessibility. Data, codebook and the entire study codebook of Korean Social Life, Health, and Aging Project are available at https://osf.io/azrsy/?view_only=d6dd38ffbd7244e6b0cf9928c deefe3c. Also, the code to replicate our analyses is available at https://github.com/JunsolKim/neural-and-social-correlates-of-attitu dinal-brokerage.

Authors' contributions. Y.Y., as the PI of the Korean Social Life, Health, and Aging Project (KSHAP), laid out and collected the survey data that produced the complete social network data of two villages. Y.Y. and J.K. designed and carried out the main fMRI and social network analyses. J.C. conceived and collected the fMRI of a sub-sample of the KSHAP respondents. J.C. and S.K. were responsible for cleaning and first-stage analyses of the fMRI data. All authors wrote and participated in revising the manuscript. Y.Y. and J.K. contributed equally to the manuscript.

Competing interests. We declare we have no competing interests.

Funding. This work was supported by the Ministry of Education of the Republic of Korea and the National Research Foundation of Korea (NRF-2017S1A3A2067165).

Acknowledgement. The authors thank Hairin Kim for her helpful feedbacks regarding the fMRI data analyses. Ekaterina Baldina provided helpful feedback on earlier drafts. Shinyoung Lee provided helpful advice on the use of Amazon Web Services.

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
