## [Reviewer comments · Proceedings of the Royal Society B: Biological Sciences]

Review History

RSPB-2020-0534.R0 (Original submission)

Review form: Reviewer 1

Recommendation

Major revision is needed (please make suggestions in comments)

Scientific importance: Is the manuscript an original and important contribution to its field?

Excellent

General interest: Is the paper of sufficient general interest?

Excellent

Quality of the paper: Is the overall quality of the paper suitable?

Good

Is the length of the paper justified?

Yes

Should the paper be seen by a specialist statistical reviewer?

Yes

Do you have any concerns about statistical analyses in this paper? If so, please specify them explicitly in your report.

Yes

It is a condition of publication that authors make their supporting data, code and materials available - either as supplementary material or hosted in an external repository. Please rate, if applicable, the supporting data on the following criteria.

Is it accessible?

Yes

Is it clear?

Yes

Is it adequate?

No

Do you have any ethical concerns with this paper?

No

Comments to the Author

In this manuscript, the Authors report the results of an fmri study that links brain network data with social network data. Specifically, the Authors collect social network data from two rural villages in South Korea. From these social networks, a subset of individuals are selected for resting state functional magnetic resonance imaging. Connectome-based predictive modeling shows that intrinsic connectivity between structures in the so-called "mentalizing" network, as well as intrinsic whole-brain connectivity, is positively predictive of measures of gender role attitude. Various cross validation and reproducible analyses demonstrate the robustness of this effect. The Authors interpret these findings as evidence demonstrating the neural basis of attitudinal brokerage processes.

There is a lot to like about this study. The sample alone is quite unique and the data analyses are sophisticated. I commend the Authors for pulling off a large sample neuroimaging study that makes considerable progress in advancing our understanding of how the brain facilitates interactions in social networks. With that said, I do have a number of questions for the Authors. I hope these questions help the Authors as they seek to revise their manuscript.

Main Comments:

1. The rationale for including mentalizing, both structurally and functionally, needs expanded on. After reading the Authors argumentation on pp. 1-2, I have three concerns. First, a more clear demonstration of how the findings in citations 38 - 45 support the definition of mentalizing would be useful. While I agree that structures in the mentalizing network are likely implicated in attitudinal brokerage processes, I found the argumentation for why a little thin and not completely convincing. A second, and related concern, is that many of the citations the Authors include as implicated in mentalizing are task evoked. Why is it that we should expect that intrinsic connectivity during rest, particularly within structures commonly implicated in mentalizing, is predictive of attitudinal brokerage? This gets at my final concern about the rationale for the hypothesis articulated on p.2 (lines 54- 56). This hypothesis relies on a reverse inference for what structures in the mentalizing network are "doing" during rest. However, excluding the TPJ, the dmPFC and precuneus are also implicated in the default mode network. Why is it, especially during rest, that connectivity between these structures is indicative of mentalizing processes, and not more general intrinsic connectivity associated with the brain at rest (this is where the Authors' whole-brain CPM actually works against them - see comment #2 below)? Since this is a core justification for a major hypothesis and analytic decision in this study,

more detail and justification would help.

2. My lab does not use the toolboxes described in this paper for network construction, so I am not very familiar with them. To make sure I understand what the Authors did when making their adjacency matrices: pre-processed neural data were subject to nuisance regression before adjacency matrix construction. Nuisance regression included the following parameters: six motion parameters, parameters for CSF and WM, and nuisance parameters for outlier volumes (global mean intensity Z-values > 5 and movement > 0.9 mm). Do I understand correctly?

I ask because I am trying to compare what the Authors did to recent bench-marking for cleaning functional connectivity data. A recent paper by Ciric and colleagues (<https://doi.org/10.1016/j.neuroimage.2017.03.020>) bench-marked a number of cleaning pipelines, and provides a number of data-driven suggestions. As far as I can tell, the pipeline reported in this study is a hybrid of what that Ciric et al. call the 2p, 6p, and spike regression cleaning pipelines. One of the major conclusions of the Ciric et al. paper is that simple regression strategies do not adequately remove motion confounds. What I can't decide is how effective the cleaning pipeline reported in this study is. On one hand, the pipeline does not include global signal regression (GSR; which uniformly does better than pipelines that do not include GSR). On the other hand, it looks like the cleaning pipeline is similar to the 9p model + models including spike regression, both of which performed well in the Ciric et al. benchmarks. While there certainly is no "gold standard" cleaning pipeline (even the Ciric paper argues this), it does seem like at least some procedures are gaining prominence (e.g., Ciric et al argue for either AROMA + GSR or 36p + censoring procedures + GSR depending on analytical goals).

My question is, why is the Authors' pipeline appropriate? One of the core concerns is that, when an inadequate cleaning pipeline is applied, edges in a network are correlated with motion (Ciric et al., Figure 2), distance (Ciric et al., Figure 4), and overall quality of the cleaning pipeline (Ciric et al., Figure 3). In some ways, that the whole-brain CPM corroborates the mentalizing results makes me worried that what the Authors are reporting might actually be driven by nothing more than a motion artifact.

3. I'm not familiar with the constraint metric the Authors propose. My understanding of what the Authors mean when they conceptually define attitudinal brokerage is the equivalent of a hub node in a network. There certainly are plenty of ways of defining a hub node, be it eigenvalue centrality, betweenness/closeness centrality, high delta centrality, etc. These measures, often are highly correlated. Why is the constraint metric most appropriate for identifying hub nodes in this study?

4. The ambivalence score is bimodal (figure S2). Does that create a problem for the Authors' CPM?

5. The abstract implicates both mentalizing and moral judgment networks in predicting social network data. However, the introduction and rationale for the study only discusses mentalizing. The disconnect is a little odd. Related, the treatment of moral correlates in the discussion section is rather thin and post-hoc. I would prefer if the Authors remove it. Instead, a focus on mentalizing and related social cognition processes is probably most appropriate in the discussion section.

Minor comments:

1. Is figure S1a mislabeled? Figure S1a is suggested to implicate an individual with high brokerage. However, the way brokerage is commonly conceptualized is a node that connects two different subgraphs or small worlds. Even more simply, a brokerage node is a hub node. In this paper, the Authors describe attitudinal brokerage, which is an individual that connects two otherwise disconnected subgraphs, which is appropriate. But that is not what is reflected in

figure S1a. Seems to me that S1a describes network closure, whereas S1b describes a hub node.

2. When drawing edges between nodes, did both individuals have to indicate a connection in the survey data? Only one individual?

3. What is the reliability estimate of the two items in the gender roles instrument? Given that it is just two items, reporting a Spearman-Brown coefficient is probably best (see: <https://doi.org/10.1007/s00038-012-0416-3>)

4. If gender role attitudes is the average of responses to two 5-point likert scales, then the midpoint would be 2.5. Why is it that, in equation one (ambivalence), egalitarian social network members are defined as those who scored 3, and not simply above the midpoint, on the gender role attitudes measure?

5. On p.9, line 228, the Authors write: "To complement this theory-driven approach..." Saying "theory driven" seems a bit strong here. I'm willing to believe that the Authors conduct a hypothesis driven analysis. But I see nothing in the paper's rationale that implies a strong theory (see main comment #1).

6. I see that the Authors make their raw data available. Is there any reason the code is also not available?

Review form: Reviewer 2

Recommendation

Major revision is needed (please make suggestions in comments)

Scientific importance: Is the manuscript an original and important contribution to its field?

Excellent

General interest: Is the paper of sufficient general interest?

Excellent

Quality of the paper: Is the overall quality of the paper suitable?

Good

Is the length of the paper justified?

Yes

Should the paper be seen by a specialist statistical reviewer?

Yes

Do you have any concerns about statistical analyses in this paper? If so, please specify them explicitly in your report.

No

It is a condition of publication that authors make their supporting data, code and materials available - either as supplementary material or hosted in an external repository. Please rate, if applicable, the supporting data on the following criteria.

Is it accessible?

Yes

Is it clear?

No

Is it adequate?

No

Do you have any ethical concerns with this paper?

No

Comments to the Author

This paper reports on an impressive dataset in which the team recorded attitudinal information (about gender attitudes) from nearly all of the elders in two Korean villages and scanned the brains of a subset of these participants with fMRI. The dataset is unique and the questions the authors have asked are important. The results are intriguing. That said, several additional clarifications and analyses would strengthen the paper.

Conceptual questions:

Prior work suggests that brain activity and connectivity in the mentalizing system are associated with social network position. In this dataset, it isn't clear whether that is the case. Is connectivity in the mentalizing system related to their measure of brokerage or other network position variables? This seems directly relevant to the interpretation of the brain-attitudinal diversity results, as well as the brain*brokerage interaction.

It also seems important to know whether the participant's own baseline attitudes relate to their network's attitudinal diversity. For example, someone with a moderate attitude (in the middle of the scale) might have more ability to connect with different positions, and this might also moderate the mentalizing \square ambivalence/SD prediction, or someone with more extreme positions might look like they have more diversity because they are bridging moderate and extreme views. Can you clarify whether a participant's own attitudes are related to any of the key predictors or outcomes and whether the main results hold when controlling for the participant's own attitudes?

Related to the point above, in the discussion, you note that there are two kinds of computations to resolve (differences between my own and others' attitudes and then differences between friends attitudes) but this isn't really tested empirically. It seems like the data you have would allow you to separate this, for example by controlling for participants' own attitudes and showing that the relationships go above and beyond their own initial position, or by directly focusing on whether the same neural processes are associated with distance from the participants' own attitudes to those of their connections vs. the diversity of the network per se. The authors may also have other ways of addressing this point, but I thought their discussion of it was relevant and that this could benefit from empirical investigation.

Greater clarification of the theoretical argument linking the brain activity to the network variables would help. For example, the authors argue in support of their observed interaction between mentalizing connectivity and brokerage "he or she still needs to be surrounded by people with different attitudes to exercise that neural capacity" - but an alternative argument would be that people who have certain neural tendencies might seek out different views, or that people who are surrounded by people with different views might develop different mentalizing tendencies. I think it would be helpful to understand whether different model specifications with these same variables give any insight into whether they are all equally plausible or whether the specific order proposed by the authors is most likely given the data (i.e., if you test a model where mentalizing is predicted by an interaction between attitudinal diversity and brokerage, is the fit worse than when mentalizing is the predictor? What about if you look at the interaction between mentalizing and attitudinal variables on brokerage? Etc.). This might also help clarify the argument. Related to this in some places the the authors seem to frame the logic like a mediation

(network-brain-attitudes) but then test moderation of attitudes \sim brain*network position.

I appreciated the authors use of open science practices such as posting their data on OSF. I apologize if I missed it, but I didn't see a protocol document/ full study code book or place to see what else was collected and to get more detail about the full measures that produced the data. Can the authors post such a list (of all the measures collected)? Since there is no pre-registration it is difficult to tell whether the attitudinal measure might be the only proxy they have for attitudinal diversity or whether other related variables should be checked. Specifically, the authors argue that resting connectivity within the mentalizing system is associated with attitudinal diversity of participants' social networks, but this is inferred from only one kind of attitude (i.e., attitudes about gender). Do you have any other types of attitudes that were measured where you can see if these effects are robust for different kinds of attitudes? If not, the discussion should acknowledge this limitation. Please also add in summary info in data that are available on OSF to make reproducibility easier (e.g., a study code book). It would also be helpful to include analysis code as well so that it is clear how the data produce the outcomes reported.

If the whole brain connectivity is a significant predictor of the outcome, what does that mean/ can we conclude anything about mentalizing specifically? What if you take random subsets of nodes that are the same size as the mentalizing network? Is mentalizing above what you'd get by chance? Does mentalizing connectivity predict above and beyond the whole brain connectivity?

Methods clarifications:

The authors note that they defined the ROI using neurosynth from studies that frequently used the term "mentalizing," but don't specify which type of map on neurosynth. Please provide more detail about the ROI definition.

How was the sub-population of scanned participants selected? In the supplemental materials you note some exclusion criteria, but was everyone who didn't have cognitive impairment or the criteria you screened for scanned?

Social network size was measured as the number of people connected to each participant in the village-wise complete social networks. Is that based on who they nominated, who nominated them, both, something else?

Minor:

The text "including six degrees of separation between people in the United States" was a little confusing to me. It makes it seem like there are 6 degrees of separation between the topics that follows in the list too. Consider re-phrasing.

The authors argue that their work is relevant to "preventing political polarization" \square I might frame as "reducing" since this work seems unlikely to be able to fully prevent polarization. When introducing ambivalence, please give intuition about what higher and lower values would mean, what the range is, etc.

What is negative prediction and positive prediction? Do you mean edges in the connectivity graph that are positively connected vs. negatively connected? For example, where you write "On the other hand, neither ambivalence scores ($r = 0.1136$, $p = 0.265$; $MAE = 0.8971$, $p = 0.274$) nor standard deviations ($r = -0.0182$, $p = 0.557$; $MAE = 0.9232$, $p = 0.712$) were negatively predicted by mentalizing brain connectivity." how does this analysis differ from the one in the prior sentence about positive prediction? Why not include both types of edges in the same model to predict outcomes/ why separate them?

For the analysis showing that connectivity in mentalizing regions from one village works in the second village, was this using the exact weights defined in one village or just the same edges but updating the weights?

For the social correlates section, at first I wondered why the authors dichotomize everything and throw away the continuous info? Why not use continuous mentalizing and brokerage scores? Later, the authors report verifying this with a continuous measure but not clear what they found: "Occupation of a brokerage position could moderate the association between mentalizing brain functional connectivity and ambivalence scores (see Supplementary Information text)." - what does this mean? I wasn't sure which part of the supplementary text addressed this directly. Can you add another sentence or two in the main manuscript to clarify?

You noted motion parameters were included as nuisance regressors. How many? 6? 12? 24? Did you regress out any physiological noise?

Sp7, line 149, pval missing a decimal point.

Fig S1, it looks like the labels are reversed (re A vs. B and brokerage vs. closure)
Please add more to the figure captions to say what you think the take home point is for each graph. For example, Figure S5, what is the red line? Please clarify in the figure legend or label more clearly. Fig. S7. Cluster of similar attitudes - what should the take home point for this graph be? Across all of the figures, more descriptive figure legends would be helpful. Also, please add error bars. Fig S8. What are the red vs. yellow lines in the circle plot.

Decision letter (RSPB-2020-0534.R0)

18-May-2020

Dear Dr Youm:

I am writing to inform you that your manuscript RSPB-2020-0534 entitled "Neural and social correlates of attitudinal brokerage: using the complete social networks of two entire villages" has, in its current form, been rejected for publication in Proceedings B.

This action has been taken on the advice of referees, who have recommended that substantial revisions are necessary. With this in mind we would be happy to consider a resubmission, provided the comments of the referees are fully addressed. However please note that this is not a provisional acceptance. Please clarify in your response and in the revision the ethical side of the research. Your work is based on human data and should therefore show informed consent for use in this study, as well as approval by an appropriate ethical committee.

Sincerely,
 Professor Hans Heesterbeek
 mailto: proceedingsb@royalsociety.org

Associate Editor
 Board Member: 1

Comments to Author:

Although both reviewers found your work to be a valuable contribution, they point out several issues that will need to be addressed in full. I would be glad to reconsider a revised manuscript and therefore will provide you with the opportunity to submit a revised version of this work which takes account of all the points raised by the reviewers.

Reviewer(s)' Comments to Author:

Referee: 1

Comments to the Author(s)

In this manuscript, the Authors report the results of an fmri study that links brain network data with social network data. Specifically, the Authors collect social network data from two rural villages in South Korea. From these social networks, a subset of individuals are selected for resting state functional magnetic resonance imaging. Connectome-based predictive modeling shows that intrinsic connectivity between structures in the so-called "mentalizing" network, as well as intrinsic whole-brain connectivity, is positively predictive of measures of gender role attitude. Various cross validation and reproducible analyses demonstrate the robustness of this effect. The Authors interpret these findings as evidence demonstrating the neural basis of attitudinal brokerage processes.

There is a lot to like about this study. The sample alone is quite unique and the data analyses are sophisticated. I commend the Authors for pulling off a large sample neuroimaging study that makes considerable progress in advancing our understanding of how the brain facilitates interactions in social networks. With that said, I do have a number of questions for the Authors. I hope these questions help the Authors as they seek to revise their manuscript.

Main Comments:

1. The rationale for including mentalizing, both structurally and functionally, needs expanded on. After reading the Authors argumentation on pp. 1-2, I have three concerns. First, a more clear demonstration of how the findings in citations 38 - 45 support the definition of mentalizing would be useful. While I agree that structures in the mentalizing network are likely implicated in attitudinal brokerage processes, I found the argumentation for why a little thin and not completely convincing. A second, and related concern, is that many of the citations the Authors include as implicated in mentalizing are task evoked. Why is it that we should expect that intrinsic connectivity during rest, particularly within structures commonly implicated in mentalizing, is predictive of attitudinal brokerage? This gets at my final concern about the rationale for the hypothesis articulated on p.2 (lines 54- 56). This hypothesis relies on a reverse inference for what structures in the mentalizing network are "doing" during rest. However, excluding the TPJ, the dmPFC and precuneus are also implicated in the default mode network. Why is it, especially during rest, that connectivity between these structures is indicative of mentalizing processes, and not more general intrinsic connectivity associated with the brain at rest (this is where the Authors' whole-brain CPM actually works against them - see comment #2 below)? Since this is a core justification for a major hypothesis and analytic decision in this study, more detail and justification would help.

2. My lab does not use the toolboxes described in this paper for network construction, so I am not very familiar with them. To make sure I understand what the Authors did when making their adjacency matrices: pre-processed neural data were subject to nuisance regression before adjacency matrix construction. Nuisance regression included the following parameters: six motion parameters, parameters for CSF and WM, and nuisance parameters for outlier volumes (global mean intensity Z-values > 5 and movement > 0.9 mm). Do I understand correctly?

I ask because I am trying to compare what the Authors did to recent bench-marking for cleaning functional connectivity data. A recent paper by Ciric and colleagues (<https://doi.org/10.1016/j.neuroimage.2017.03.020>) bench-marked a number of cleaning pipelines, and provides a number of data-driven suggestions. As far as I can tell, the pipeline reported in this study is a hybrid of what that Ciric et al. call the 2p, 6p, and spike regression cleaning pipelines. One of the major conclusions of the Ciric et al. paper is that simple regression strategies do not adequately remove motion confounds. What I can't decide is how effective the cleaning pipeline reported in this study is. On one hand, the pipeline does not include global signal regression (GSR; which uniformly does better than pipelines that do not include GSR). On the other hand, it looks like the cleaning pipeline is similar to the 9p model + models including spike regression, both of which performed well in the Ciric et al. benchmarks. While there certainly is no "gold standard" cleaning pipeline (even the Ciric paper argues this), it does seem like at least some procedures are gaining prominence (e.g., Ciric et al argue for either AROMA + GSR or 36p + censoring procedures + GSR depending on analytical goals).

My question is, why is the Authors' pipeline appropriate? One of the core concerns is that, when an inadequate cleaning pipeline is applied, edges in a network are correlated with motion (Ciric et al., Figure 2), distance (Ciric et al., Figure 4), and overall quality of the cleaning pipeline (Ciric et al., Figure 3). In some ways, that the whole-brain CPM corroborates the mentalizing results makes me worried that what the Authors are reporting might actually be driven by nothing more than a motion artifact.

3. I'm not familiar with the constraint metric the Authors propose. My understanding of what the Authors mean when they conceptually define attitudinal brokerage is the equivalent of a hub node in a network. There certainly are plenty of ways of defining a hub node, be it eigenvalue centrality, betweenness/closeness centrality, high delta centrality, etc. These measures, often are highly correlated. Why is the constraint metric most appropriate for identifying hub nodes in this study?

4. The ambivalence score is bimodal (figure S2). Does that create a problem for the Authors' CPM?

5. The abstract implicates both mentalizing and moral judgment networks in predicting social network data. However, the introduction and rationale for the study only discusses mentalizing. The disconnect is a little odd. Related, the treatment of moral correlates in the discussion section is rather thin and post-hoc. I would prefer if the Authors remove it. Instead, a focus on mentalizing and related social cognition processes is probably most appropriate in the discussion section.

Minor comments:

1. Is figure S1a mislabeled? Figure S1a is suggested to implicate an individual with high brokerage. However, the way brokerage is commonly conceptualized is a node that connects two different subgraphs or small worlds. Even more simply, a brokerage node is a hub node. In this paper, the Authors describe attitudinal brokerage, which is an individual that connects two otherwise disconnected subgraphs, which is appropriate. But that is not what is reflected in figure S1a. Seems to me that S1a describes network closure, whereas S1b describes a hub node.

2. When drawing edges between nodes, did both individuals have to indicate a connection in the survey data? Only one individual?

3. What is the reliability estimate of the two items in the gender roles instrument? Given that it is just two items, reporting a Spearman-Brown coefficient is probably best (see: <https://doi.org/10.1007/s00038-012-0416-3>)

4. If gender role attitudes is the average of responses to two 5-point likert scales, then the midpoint would be 2.5. Why is it that, in equation one (ambivalence), egalitarian social network members are defined as those who scored 3, and not simply above the midpoint, on the gender role attitudes measure?

5. On p.9, line 228, the Authors write: "To complement this theory-driven approach..." Saying "theory driven" seems a bit strong here. I'm willing to believe that the Authors conduct a hypothesis driven analysis. But I see nothing in the paper's rationale that implies a strong theory (see main comment #1).

6. I see that the Authors make their raw data available. Is there any reason the code is also not available?

Referee: 2

Comments to the Author(s)

This paper reports on an impressive dataset in which the team recorded attitudinal information (about gender attitudes) from nearly all of the elders in two Korean villages and scanned the brains of a subset of these participants with fMRI. The dataset is unique and the questions the authors have asked are important. The results are intriguing. That said, several additional clarifications and analyses would strengthen the paper.

Conceptual questions:

Prior work suggests that brain activity and connectivity in the mentalizing system are associated with social network position. In this dataset, it isn't clear whether that is the case. Is connectivity in the mentalizing system related to their measure of brokerage or other network position variables? This seems directly relevant to the interpretation of the brain-attitudinal diversity results, as well as the brain*brokerage interaction.

It also seems important to know whether the participant's own baseline attitudes relate to their network's attitudinal diversity. For example, someone with a moderate attitude (in the middle of the scale) might have more ability to connect with different positions, and this might also moderate the mentalizing \square ambivalence/SD prediction, or someone with more extreme positions might look like they have more diversity because they are bridging moderate and extreme views. Can you clarify whether a participant's own attitudes are related to any of the key predictors or outcomes and whether the main results hold when controlling for the participant's own attitudes?

Related to the point above, in the discussion, you note that there are two kinds of computations to resolve (differences between my own and others' attitudes and then differences between friends attitudes) but this isn't really tested empirically. It seems like the data you have would allow you to separate this, for example by controlling for participants' own attitudes and showing that the relationships go above and beyond their own initial position, or by directly focusing on whether the same neural processes are associated with distance from the participants' own attitudes to those of their connections vs. the diversity of the network per se. The authors may also have other ways of addressing this point, but I thought their discussion of it was relevant and that this could benefit from empirical investigation.

Greater clarification of the theoretical argument linking the brain activity to the network variables would help. For example, the authors argue in support of their observed interaction between mentalizing connectivity and brokerage "he or she still needs to be surrounded by people with

different attitudes to exercise that neural capacity” – but an alternative argument would be that people who have certain neural tendencies might seek out different views, or that people who are surrounded by people with different views might develop different mentalizing tendencies. I think it would be helpful to understand whether different model specifications with these same variables give any insight into whether they are all equally plausible or whether the specific order proposed by the authors is most likely given the data (i.e., if you test a model where mentalizing is predicted by an interaction between attitudinal diversity and brokerage, is the fit worse than when mentalizing is the predictor? What about if you look at the interaction between mentalizing and attitudinal variables on brokerage? Etc.). This might also help clarify the argument. Related to this in some places the authors seem to frame the logic like a mediation (network-brain-attitudes) but then test moderation of attitudes ~ brain*network position.

I appreciated the authors use of open science practices such as posting their data on OSF. I apologize if I missed it, but I didn't see a protocol document/ full study code book or place to see what else was collected and to get more detail about the full measures that produced the data.

Can the authors post such a list (of all the measures collected)? Since there is no pre-registration it is difficult to tell whether the attitudinal measure might be the only proxy they have for attitudinal diversity or whether other related variables should be checked. Specifically, the authors argue that resting connectivity within the mentalizing system is associated with attitudinal diversity of participants' social networks, but this is inferred from only one kind of attitude (i.e., attitudes about gender). Do you have any other types of attitudes that were measured where you can see if these effects are robust for different kinds of attitudes? If not, the discussion should acknowledge this limitation. Please also add in summary info in data that are available on OSF to make reproducibility easier (e.g., a study code book). It would also be helpful to include analysis code as well so that it is clear how the data produce the outcomes reported.

If the whole brain connectivity is a significant predictor of the outcome, what does that mean/ can we conclude anything about mentalizing specifically? What if you take random subsets of nodes that are the same size as the mentalizing network? Is mentalizing above what you'd get by chance? Does mentalizing connectivity predict above and beyond the whole brain connectivity?

Methods clarifications:

The authors note that they defined the ROI using neurosynth from studies that frequently used the term “mentalizing,” but don't specify which type of map on neurosynth. Please provide more detail about the ROI definition.

How was the sub-population of scanned participants selected? In the supplemental materials you note some exclusion criteria, but was everyone who didn't have cognitive impairment or the criteria you screened for scanned?

Social network size was measured as the number of people connected to each participant in the village-wise complete social networks. Is that based on who they nominated, who nominated them, both, something else?

Minor:

The text “including six degrees of separation between people in the United States” was a little confusing to me. It makes it seem like there are 6 degrees of separation between the topics that follows in the list too. Consider re-phrasing.

The authors argue that their work is relevant to “preventing political polarization” □ I might frame as “reducing” since this work seems unlikely to be able to fully prevent polarization. When introducing ambivalence, please give intuition about what higher and lower values would mean, what the range is, etc.

What is negative prediction and positive prediction? Do you mean edges in the connectivity graph that are positively connected vs. negatively connected? For example, where you write “On

the other hand, neither ambivalence scores ($r = 0.1136$, $p = 0.265$; $MAE = 0.8971$, $p = 0.274$) nor standard deviations ($r = -0.0182$, $p = 0.557$; $MAE = 0.9232$, $p = 0.712$) were negatively predicted by mentalizing brain connectivity." how does this analysis differ from the one in the prior sentence about positive prediction? Why not include both types of edges in the same model to predict outcomes/ why separate them?

For the analysis showing that connectivity in mentalizing regions from one village works in the second village, was this using the exact weights defined in one village or just the same edges but updating the weights?

For the social correlates section, at first I wondered why the authors dichotomize everything and throw away the continuous info? Why not use continuous mentalizing and brokerage scores?

Later, the authors report verifying this with a continuous measure but not clear what they found: "Occupation of a brokerage position could moderate the association between mentalizing brain functional connectivity and ambivalence scores (see Supplementary Information text)." - what does this mean? I wasn't sure which part of the supplementary text addressed this directly. Can you add another sentence or two in the main manuscript to clarify?

You noted motion parameters were included as nuisance regressors. How many? 6? 12? 24? Did you regress out any physiological noise?

Sp7, line 149, pval missing a decimal point.

Fig S1, it looks like the labels are reversed (re A vs. B and brokerage vs. closure)

Please add more to the figure captions to say what you think the take home point is for each graph. For example, Figure S5, what is the red line? Please clarify in the figure legend or label more clearly. Fig. S7. Cluster of similar attitudes - what should the take home point for this graph be? Across all of the figures, more descriptive figure legends would be helpful. Also, please add error bars. Fig S8. What are the red vs. yellow lines in the circle plot.

Author's Response to Decision Letter for (RSPB-2020-0534.R0)

See Appendix A.

RSPB-2020-2866.R0

Review form: Reviewer 1

Recommendation

Accept as is

Scientific importance: Is the manuscript an original and important contribution to its field?

Excellent

General interest: Is the paper of sufficient general interest?

Excellent

Quality of the paper: Is the overall quality of the paper suitable?

Excellent

Is the length of the paper justified?

Yes

Should the paper be seen by a specialist statistical reviewer?

No

Do you have any concerns about statistical analyses in this paper? If so, please specify them explicitly in your report.

No

It is a condition of publication that authors make their supporting data, code and materials available - either as supplementary material or hosted in an external repository. Please rate, if applicable, the supporting data on the following criteria.

Is it accessible?

Yes

Is it clear?

Yes

Is it adequate?

Yes

Do you have any ethical concerns with this paper?

No

Comments to the Author

I would like to thank the Authors for their detailed response letter. The additional clarification and analysis resolves nearly all of my concerns. The low reliability value for the gender roles instrument, especially given the critical role of this measure, certainly is a concern that constrains our interpretation of the results. However, and given that this measure has been used in previous research, I do see some value in reusing the same measure in the present study. It is also worth noting that the Authors are transparent about this potential limitation, which ultimately leaves it up to the reader to decide when evaluating this manuscript. I think this is appropriate.

I believe the Authors have produced an impressive manuscript. My concerns are resolved.

Review form: Reviewer 2

Recommendation

Accept with minor revision (please list in comments)

Scientific importance: Is the manuscript an original and important contribution to its field?

Excellent

General interest: Is the paper of sufficient general interest?

Excellent

Quality of the paper: Is the overall quality of the paper suitable?

Good

Is the length of the paper justified?

Yes

Should the paper be seen by a specialist statistical reviewer?

Yes

Do you have any concerns about statistical analyses in this paper? If so, please specify them explicitly in your report.

No

It is a condition of publication that authors make their supporting data, code and materials available - either as supplementary material or hosted in an external repository. Please rate, if applicable, the supporting data on the following criteria.

Is it accessible?

Yes

Is it clear?

Yes

Is it adequate?

Yes

Do you have any ethical concerns with this paper?

No

Comments to the Author

The authors did a nice job revising the paper and clarified several of my previous questions. I have a few more suggestions/ clarifications before publication:

The authors used “lesioned connectivity vs mentalizing connectivity” as a way to show that effects are specific to mentalizing. Why not include both in the same model and show that mentalizing explains above and beyond the rest of the brain (i.e.. above and beyond the lesioned connectivity)? Also, when comparing the correlations, there doesn’t seem to be a direct comparison/ test of the difference between the r values. It would be good to test this formally.

Related to the question above, several of the tables report both correlation and mean absolute error. You do a nice job explaining to the reader what each represents. Can you add one more sentence to explain to the reader how to interpret differences (e.g., when one is significant and the other isn’t)?

The authors write: “As shown in Table S7 and Fig. S11, occupation of a structural brokerage position significantly moderated the association between mentalizing brain functional connectivity and ambivalence scores ($p=0.057$).” But I think significant here means $p<0.05$. Consider revising to maybe “marginally moderated” or something like that?

Fig S3 – The legend says “attitudinal diversity” and y axis says ambivalence. Remind the reader that ambivalence is one operationalization of diversity.

The authors used two items re: gender attitudes, and noted: “In our study, the reliability estimate of the two items was not so high (Spearman-Brown coefficient = 0.222).” – are the results more strongly driven by one item than the other? If they are capturing different things, it might be useful to have a supplemental analysis that shows how the main results look if you use each item separately, and if one is driving the results, briefly discuss in the discussion.

The authors deleted “Additional studies examining younger populations and/or urban areas in countries with different cultural and social contexts would be helpful to confirm the generality of our findings. The source code to estimate the model to predict attitudinal diversity of social network members employed in this study, along with the corresponding dataset, is available for the purpose of replication.” - I think this text would be useful to keep.

Please add code to OSF page—the data are nicely organized and clear, but I didn’t easily find code to reproduce the analyses/results.

Decision letter (RSPB-2020-2866.R0)

29-Dec-2020

Dear Dr Youm:

Your manuscript has now been peer reviewed and the reviews have been assessed by an Associate Editor. The reviewers’ comments (not including confidential comments to the Editor) and the comments from the Associate Editor are included at the end of this email for your reference. As you will see, the reviewers have raised some issues with your manuscript and we would like to invite you to revise your manuscript to address them.

When submitting your revision please upload a file under "Response to Referees" in the "File Upload" section. This should document, point by point, how you have responded to the reviewers’ and Editors’ comments, and the adjustments you have made to the manuscript. We require a copy of the manuscript with revisions made since the previous version marked as ‘tracked changes’ to be included in the ‘response to referees’ document.

Research ethics:

Use of animals and field studies:

If your study uses animals please include details in the methods section of any approval and licences given to carry out the study and include full details of how animal welfare standards

were ensured. Field studies should be conducted in accordance with local legislation; please include details of the appropriate permission and licences that you obtained to carry out the field work.

It is a condition of publication that you make available the data and research materials supporting the results in the article (<https://royalsociety.org/journals/authors/author-guidelines/#data>). Datasets should be deposited in an appropriate publicly available repository and details of the associated accession number, link or DOI to the datasets must be included in the Data Accessibility section of the article (<https://royalsociety.org/journals/ethics-policies/data-sharing-mining/>). Reference(s) to datasets should also be included in the reference list of the article with DOIs (where available).

If you wish to submit your data to Dryad (<http://datadryad.org/>) and have not already done so you can submit your data via this link [http://datadryad.org/submit?journalID=RSPB&manu=\(Document not available\)](http://datadryad.org/submit?journalID=RSPB&manu=(Document%20not%20available)), which will take you to your unique entry in the Dryad repository.

Please submit a copy of your revised paper within three weeks. If we do not hear from you within this time your manuscript will be rejected. If you are unable to meet this deadline please let us know as soon as possible, as we may be able to grant a short extension.

Best wishes,
Professor Hans Heesterbeek
<mailto:proceedingsb@royalsociety.org>

Associate Editor Board Member

Comments to Author:

Both reviewers agree that the manuscript has substantially improved and potentially meets the standard for publication in Proc B. However, we would like you to make some final revisions to your manuscript in response to the suggestions from reviewer 2. These suggestions mainly

comprise several clarifications, but also include an additional test to demonstrate the statistical difference among r values.

Reviewer(s)' Comments to Author:

Referee: 1

Comments to the Author(s).

I would like to thank the Authors for their detailed response letter. The additional clarification and analysis resolves nearly all of my concerns. The low reliability value for the gender roles instrument, especially given the critical role of this measure, certainly is a concern that constrains our interpretation of the results. However, and given that this measure has been used in previous research, I do see some value in reusing the same measure in the present study. It is also worth noting that the Authors are transparent about this potential limitation, which ultimately leaves it up to the reader to decide when evaluating this manuscript. I think this is appropriate.

I believe the Authors have produced an impressive manuscript. My concerns are resolved.

Referee: 2

Comments to the Author(s).

The authors did a nice job revising the paper and clarified several of my previous questions. I have a few more suggestions/ clarifications before publication:

The authors used “lesioned connectivity vs mentalizing connectivity” as a way to show that effects are specific to mentalizing. Why not include both in the same model and show that mentalizing explains above and beyond the rest of the brain (i.e., above and beyond the lesioned connectivity)? Also, when comparing the correlations, there doesn’t seem to be a direct comparison/ test of the difference between the r values. It would be good to test this formally.

Related to the question above, several of the tables report both correlation and mean absolute error. You do a nice job explaining to the reader what each represents. Can you add one more sentence to explain to the reader how to interpret differences (e.g., when one is significant and the other isn’t)?

The authors write: “As shown in Table S7 and Fig. S11, occupation of a structural brokerage position significantly moderated the association between mentalizing brain functional connectivity and ambivalence scores ($p=0.057$).” But I think significant here means $p<.05$. Consider revising to maybe “marginally moderated” or something like that?

Fig S3 – The legend says “attitudinal diversity” and y axis says ambivalence. Remind the reader that ambivalence is one operationalization of diversity.

The authors used two items re: gender attitudes, and noted: “In our study, the reliability estimate of the two items was not so high (Spearman-Brown coefficient = $=0.222$).” – are the results more strongly driven by one item than the other? If they are capturing different things, it might be useful to have a supplemental analysis that shows how the main results look if you use each item separately, and if one is driving the results, briefly discuss in the discussion.

The authors deleted “Additional studies examining younger populations and/or urban areas in countries with different cultural and social contexts would be helpful to confirm the generality of our findings. The source code to estimate the model to predict attitudinal diversity of social network members employed in this study, along with the corresponding dataset, is available for the purpose of replication.” – I think this text would be useful to keep.

Please add code to OSF page—the data are nicely organized and clear, but I didn't easily find code to reproduce the analyses/results.

Author's Response to Decision Letter for (RSPB-2020-2866.R0)

See Appendix B.

Decision letter (RSPB-2020-2866.R1)

18-Jan-2021

Dear Dr Youm

I am pleased to inform you that your manuscript entitled "Neural and social correlates of attitudinal brokerage: using the complete social networks of two entire villages" has been accepted for publication in Proceedings B.

Open Access

Paper charges

Sincerely,
Professor Hans Heesterbeek
Editor, Proceedings B
mailto: proceedingsb@royalsociety.org

Appendix A

Response to Referees (No. RSPB-2020-0534)

Title: Neural and social correlates of attitudinal brokerage: using the complete social networks of two entire villages

November 17, 2020

Dear Editor,

We truly appreciate you for securing valuable reviews. We are also grateful for the helpful comments provided by both reviewers. We have revised the manuscript in the light of the comments that we received and believe that the manuscript has been strengthened as a result of incorporating this helpful feedback. In particular, as detailed in our point-by-point responses to reviewers' comments, we now showed that our key findings are robust to different data analytic choices and we have also incorporated suggestions from both reviewers to refine and deepen our interpretations of the results.

Please note that our work is based on human data and should therefore show informed consent for use in this study, as well as approval by an appropriate ethical committee. As clarified in "Acquisition of resting-state fMRI data and image processing" section in our main manuscript, the study was approved by and performed in accordance with the relevant guidelines and regulations by the Institutional Review Board of Yonsei University (IRB number: YUIRB-2011-012-01; Township K survey, brain fMRI, and Township L survey) and the Institutional Review Board of Seoul National University (IRB number: 1801/001-003; Township L brain fMRI), and all participants provided written informed consent for the research procedure.

Referee 1:

In this manuscript, the Authors report the results of an fmri study that links brain network data with social network data. Specifically, the Authors collect social network data from two rural villages in South Korea. From these social networks, a subset of individuals are selected for resting state functional magnetic resonance imaging. Connectome-based predictive modeling shows that intrinsic connectivity between structures in the so-called "mentalizing" network, as well as intrinsic whole-brain connectivity, is positively predictive of measures of gender role attitude. Various cross validation and reproducible analyses demonstrate the robustness of this effect. The Authors interpret these findings as evidence demonstrating the neural basis of attitudinal brokerage processes.

There is a lot to like about this study. The sample alone is quite unique and the data analyses are sophisticated. I commend the Authors for pulling off a large sample neuroimaging study that makes considerable progress in advancing our understanding of how the brain facilitates interactions in social networks. With that said, I do have a number of questions for the Authors. I hope these questions help the Authors as they seek to revise their manuscript.

Main Comments:

Main Comment #1

The rationale for including mentalizing, both structurally and functionally, needs expanded on. After

reading the Authors argumentation on pp. 1-2, I have three concerns. First, a more clear demonstration of how the findings in citations 38 - 45 support the definition of mentalizing would be useful. While I agree that structures in the mentalizing network are likely implicated in attitudinal brokerage processes, I found the argumentation for why a little thin and not completely convincing. A second, and related concern, is that many of the citations the Authors include as implicated in mentalizing are task evoked. Why is it that we should expect that intrinsic connectivity during rest, particularly within structures commonly implicated in mentalizing, is predictive of attitudinal brokerage? This gets at my final concern about the rationale for the hypothesis articulated on p.2 (lines 54- 56). This hypothesis relies on a reverse inference for what structures in the mentalizing network are "doing" during rest. However, excluding the TPJ, the dmPFC and precuneus are also implicated in the default mode network. Why is it, especially during rest, that connectivity between these structures is indicative of mentalizing processes, and not more general intrinsic connectivity associated with the brain at rest (this is where the Authors' whole-brain CPM actually works against them - see comment #2 below)? Since this is a core justification for a major hypothesis and analytic decision in this study, more detail and justification would help.

Answer to Main Comment #1

As Referee #1 pointed out, our original justification was rather thin in a couple of ways. There were three sub-comments under the main comment #1.

(1) *First, a more clear demonstration of how the findings in citations 38 - 45 support the definition of mentalizing would be useful. While I agree that structures in the mentalizing network are likely implicated in attitudinal brokerage processes, I found the argumentation for why a little thin and not completely convincing.*

As you suggested, we further delineated how citations 38 – 45 support that mentalizing regions are implicated in the attitudinal brokerage process. We argued that two types of cognitive challenges should be resolved to interact with diverse others. First, one needs to deal with dissimilarity between “the self and a friend” (i.e., dyadic challenges). Second, one needs to bridge dissimilarity between “two friends of oneself” (i.e., triadic challenges). Therefore, we elaborated on how papers 38 – 45 can support that mentalizing regions are implicated in resolving each type of cognitive challenge.

Before revision: Considering the essential role of attitudinal brokerage, we examined the neural and social characteristics that might account for attitudinal brokerage, measured as the “attitudinal diversity of one’s social network members.” With regard to neural characteristics, we targeted mentalizing brain regions related to humans’ ability to infer other people’s attitudes, belief states, and intentions by incorporating various social cues (30-38); these regions include the dorsomedial prefrontal cortex (dmPFC), temporoparietal junction (TPJ), and precuneus. More specifically, we presumed that mentalizing regions are associated with attitudinal brokerage because of their association with (i) affiliation in diverse social groups (38), (ii) occupation of a large social network (39-44), (iii) processing of other people’s heterogeneous attitudes (35), (iv) processing attitudes that violate one’s previous expectations (35, 45), and (v) fewer stereotypical judgments toward others.

After revision (pp. 2, line 43): Considering the essential role of attitudinal brokerage, we examined the neural and social characteristics that might account for attitudinal brokerage, measured as the “attitudinal diversity of one’s social network members.” With regard to neural characteristics, we targeted mentalizing brain regions related to humans’ ability to infer other people’s attitudes, belief states, and intentions by incorporating various social cues (17-21); these regions include the dorsomedial prefrontal cortex (dmPFC), temporoparietal junction (TPJ), and precuneus. Mentalizing brain regions might contribute to the attitudinal brokerage by resolving two distinct cognitive challenges of coping with social network members’ diverse attitudes. First, mentalizing regions may be implicated in processing dissimilarity between “the self and a friend” (i.e., dyadic challenges) (18, 22). For example, mentalizing regions were activated when liberal people try to extrapolate conservative people’s opinions, likes, and dislikes (18). Also, mentalizing regions are known to be activated when one’s expectation about other people’s attitudes are violated (22). Second, mentalizing regions may be implicated in processing dissimilarity between “two friends of oneself” (i.e., triadic challenges). For

example, mentalizing regions are implicated in one's ability to connect with friends who are affiliated in different social groups (21), which are likely to have different attitudes.

(2) . *A second, and related concern, is that many of the citations the Authors include as implicated in mentalizing are task evoked. Why is it that we should expect that intrinsic connectivity during rest, particularly within structures commonly implicated in mentalizing, is predictive of attitudinal brokerage?*

We also introduced studies that support our hypothesis that intrinsic functional connectivity (and gray matter structure) of mentalizing brain regions at rest is implicated in attitudinal brokerage. Specifically, we added the previous task-free studies about the relationship of intrinsic functional connectivity (and gray matter density) of mentalizing regions with social network size and social network diversity. We believe that these studies are in line with the statement that brain-behavior associations are driven in part by stable trait-level variation in intrinsic functional connectivity.

After revision (pp.3, line 58): ... Many studies that implied the role of mentalizing brain regions in attitudinal brokerage are task-based studies. However, some of the task-free studies also revealed that mentalizing brain regions might be associated with the attitudinal brokerage. For example, mentalizing regions' intrinsic functional connectivity at rest was associated with the size of social networks (23, 24). Also, mentalizing regions' structure (e.g., gray matter volume, white matter integrity) was associated with the size and diversity of social networks (25-28). These studies are in line with previous literature that brain-behavior associations are driven in part by stable trait-level variation in intrinsic brain connectivity (29, 30). Therefore, we hypothesized that intrinsic brain connectivity from and to mentalizing regions would be correlated with attitudinal diversity of social network members, an indicator of attitudinal brokerage.

(3) *This gets at my final concern about the rationale for the hypothesis articulated on p.2 (lines 54- 56). This hypothesis relies on a reverse inference for what structures in the mentalizing network are "doing" during rest. However, excluding the TPJ, the dmPFC and precuneus are also implicated in the default mode network. Why is it, especially during rest, that connectivity between these structures is indicative of mentalizing processes, and not more general intrinsic connectivity associated with the brain at rest (this is where the Authors' whole-brain CPM actually works against them - see comment #2 below)?*

In the Discussion section, we mentioned that mentalizing brain regions are also implicated in default mode network as one of the limitations of our study. However, we still believe that connectivity between mentalizing regions at rest could be indicative of mentalizing processes for two reasons. First, as shown in Table S6 and S7, mentalizing regions not included in default mode network, such as pSTS and pre-SMA are part of the important hub nodes that contribute most to the prediction of attitudinal diversity (In Table S6 and S7, we indicated whether each brain region belongs to the default mode network.). Second, previous studies also showed that intrinsic functional connectivity at rest and gray matter volume of mentalizing brain regions are associated with the size and diversity of social networks.

After revision (pp. 16, line 411): Forth, mentalizing brain regions are also implicated in the default mode network. Therefore, some may want to argue that mentalizing brain connectivity is not related to mentalizing activity but indicates general intrinsic connectivity associated with the brain at rest. However, we still believe that connectivity between mentalizing regions at rest indicates mentalizing processes for two reasons. First, as shown in Table S4 and Table S5, mentalizing regions not included in the default mode network, such as pSTS and pre-SMA, are part of the important hub nodes that contribute most to the prediction of attitudinal diversity. Second, previous task-free studies also showed that intrinsic functional connectivity at rest and structure of mentalizing brain regions are associated with the size and diversity of social networks (23-28). We would benefit from a future study that elucidates whether our results are also replicated in task-based experiments.

Main Comment #2

My lab does not use the toolboxes described in this paper for network construction, so I am not very familiar with them. To make sure I understand what the Authors did when making their adjacency matrices: pre-processed neural data were subject to nuisance regression before adjacency matrix construction. Nuisance regression included the following parameters: six motion parameters, parameters for CSF and WM, and nuisance parameters for outlier volumes (global mean intensity Z-values > 5 and movement > 0.9 mm). Do I understand correctly? I ask because I am trying to compare what the Authors did to recent bench-marking for cleaning functional connectivity data. A recent paper by Ciric and colleagues (<https://doi.org/10.1016/j.neuroimage.2017.03.020>) bench-marked a number of cleaning pipelines, and provides a number of data-driven suggestions. As far as I can tell, the pipeline reported in this study is a hybrid of what that Ciric et al. call the 2p, 6p, and spike regression cleaning pipelines. One of the major conclusions of the Ciric et al. paper is that simple regression strategies do not adequately remove motion confounds. What I can't decide is how effective the cleaning pipeline reported in this study is. On one hand, the pipeline does not include global signal regression (GSR; which uniformly does better than pipelines that do not include GSR). On the other hand, it looks like the cleaning pipeline is similar to the 9p model + models including spike regression, both of which performed well in the Ciric et al. benchmarks. While there certainly is no "gold standard" cleaning pipeline (even the Ciric paper argues this), it does seem like at least some procedures are gaining prominence (e.g., Ciric et al argue for either AROMA + GSR or 36p + censoring procedures + GSR depending on analytical goals). My question is, why is the Authors' pipeline appropriate? One of the core concerns is that, when an inadequate cleaning pipeline is applied, edges in a network are correlated with motion (Ciric et al., Figure 2), distance (Ciric et al., Figure 4), and overall quality of the cleaning pipeline (Ciric et al., Figure 3). In some ways, that the whole-brain CPM corroborates the mentalizing results makes me worried that what the Authors are reporting might actually be driven by nothing more than a motion artifact.

Answer to Main Comment #2

As you pointed out well, we acknowledge that there exist two issues regarding our description of the cleaning pipeline. First, we found that the cleaning pipeline was not described in detail. We did not elaborate that preprocessed neural data were subject to the following denoising process before adjacency matrix constructions. In our denoising process, nuisance regression included the following parameters:

- Twelve motion parameters (six estimated motion parameters + six first-order temporal derivatives).
- Five principal components for cerebrospinal fluid (CSF) and five principal components for white matter based on aCompCor (anatomical component-based noise correction) method (Behzadi et al., 2007, Muschelli et al., 2014).
- Nuisance parameters for outlier volumes.

Before revision: Image preprocessing and denoising was performed using the SPM12 software (Wellcome Department of Imaging Neuroscience, Institute of Neurology, London, UK) with the Conn toolbox 18.a (<http://www.nitrc.org/projects/conn>) default preprocessing pipeline. Functional images were corrected for motion and slice time and warped into MNI standard space. Images were smoothed with an 8-mm full-width half-maximum Gaussian kernel. In addition, the Artifact Detection Tools (https://www.nitrc.org/projects/artifact_detect/) were used to identify motion and signal intensity outlier images. Images with global mean intensity Z-values > 5 and movement > 0.9 mm were identified as outlier

images. Estimated motion parameters and outlier images were used as nuisance covariates in the time-series linear regression. T1-weighted images were segmented into gray matter, white matter, and cerebrospinal fluid (CSF) and warped into MNI standard space. Signals within white matter and the CSF mask were regressed out to exclude non-gray matter BOLD signal. Band-pass temporal filtering (0.008–0.09) was applied to exclude physiological noise.

After revision (Supplementary Information: pp. 3, line 54): Image preprocessing and denoising was performed using the SPM12 software (Wellcome Department of Imaging Neuroscience, Institute of Neurology, London, UK) with the Conn toolbox 18.a (<http://www.nitrc.org/projects/conn>) default preprocessing pipeline. Functional images were corrected for motion and slice time and warped into MNI standard space. Images were smoothed with an 8-mm full-width half-maximum Gaussian kernel. In addition, the Artifact Detection Tools (https://www.nitrc.org/projects/artifact_detect/) were used to identify motion and signal intensity outlier images. Images with global mean intensity Z-values > 5 and movement > 0.9 mm were identified as outlier images. In a denoising process before adjacency matrix construction, six estimated motion parameters, six first-order temporal derivatives and outlier images were used as nuisance covariates in the time-series linear regression. T1-weighted images were segmented into gray matter, white matter, and cerebrospinal fluid (CSF) and warped into MNI standard space. Based on the aCompCor (anatomical component-based noise correction) method, five principal components each from the white-matter and cerebrospinal fluid (CSF) time series were regressed out to exclude non-gray matter BOLD signal (1, 2). Band-pass temporal filtering (0.008–0.09) was applied to exclude physiological noise.

Second, why our cleaning pipeline is appropriate was not explained in detail. Therefore, we added the benchmark results of Ciric et al., 2017 that you cited. According to Ciric et al., our method (aCompCor method + twelve motion parameters) was one of the top-performing cleaning methods which adequately removes motion confounds. According to Ciric et al., aCompCor was effective in the mitigation of residual motion: “*Somewhat to our surprise, benchmark results for aCompCor (Behzadi et al., 2007, Muschelli et al., 2014) were most similar to models that included GSR.*” We did not control for GSR because some argue that GSR can introduce artifactual biases (Murphy et al. 2009) and remove potentially meaningful neural components (Chai et al. 2012).

After revision (Supplementary Information: pp. 4, line 69): According to the recent benchmarks of denoising processes, the aCompCor method was one of the top-performing methods in the mitigation of motion confounds (3). We did not control for GSR because some argue that GSR may introduce artifactual biases and remove potentially meaningful neural components (4, 5). For each subject, mean time series were extracted by averaging all voxels composing each region for each time point from the 227 regions of Shen’s whole-brain parcellation atlas (37 cerebellar regions were excluded) (6). Pearson’s correlation coefficients were calculated between each pair of regions and transformed to Fisher’s Z-scores. Therefore, 139 individual whole-brain connectivity matrices containing 25,651 ($= (227 \times (227-1))/2$) edges were constructed.

Furthermore, in order to show that this paper’s results are not a motion artifact, we also controlled for head motions in our control analyses. In particular, we additionally controlled for maximum frame-wise displacement and mean frame-wise displacement in the control analyses. Shen et al., 2017 argued that head motion should be not only corrected in cleaning pipeline but also included in control analyses. It is because head motions could artifactually increase prediction performance if head motion and outcome of interest are correlated. The results show that the prediction performance of mentalizing networks are significant even after controlling for head motions.

After revision (Supplementary Information: pp. 8, line 186): In addition to the confounders above, the head motion may cause spurious patterns of brain functional connectivity. The head motions could artifactually increase prediction performance if head motion and attitudinal diversity scores are correlated (9). Therefore, some argue that head motion should be not only corrected in the cleaning pipeline but also included in control analyses (9). In turn, we controlled for maximum frame-wise displacement and mean frame-wise displacement.

Main Comment #3

I'm not familiar with the constraint metric the Authors propose. My understanding of what the Authors mean when they conceptually define attitudinal brokerage is the equivalent of a hub node in a network. There certainly are plenty of ways of defining a hub node, be it eigenvalue centrality, betweenness/closeness centrality, high delta centrality, etc. These measures, often are highly correlated. Why is the constraint metric most appropriate for identifying hub nodes in this study?

Answer to Main Comment #3

We admit that the terms we used in the first draft could be misleading. “Brokerage position” that was measured by constraint metric and “attitudinal brokerage” are two different concepts. “Attitudinal brokerage” refers keeping connections with attitudinally different people at the same time. On the other hand, “brokerage position” means maintaining connections with otherwise unconnected people (See the Figure below). These two terms were confusing. Therefore, instead of the term “brokerage position”, we used the equivalent term “structural brokerage position” in the revised manuscript to minimize the confusion.

Also, in Sociology, “structural brokerage position” (brokerage position) is a quite different concept from “hub node”. While structural brokerage refers to a node having connections with two people who are not otherwise connected, a hub node means a node with many connections or a node that occupies a central position in a network. Therefore, one can be in a structural brokerage position without being a hub node. We modified Figure S1 to clarify the differences between structural closure and structural brokerage. We also modified Figure S1 so that readers would not confuse structural brokerage with a hub node. Furthermore, we added Table S2 showing the correlations between structural brokerage measure (i.e., structural constraint) and hub node measures (e.g., *betweenness centrality*, *closeness centrality*, and *eigenvector centrality*). The correlation coefficients were modest (between -0.636 and -0.188), which implicates that a structural brokerage position and a hub node are conceptually different.

After revision (pp. 4, line 93): It should be noted that a structural brokerage position is conceptually different from a hub node. While a node in a structural brokerage position refers to a node having connections with two people who are not otherwise connected, a hub node indicates a node having many connections or a node that occupies a central position in a network. Therefore, one can be in a structural brokerage position without being a hub node. Table S6 presents correlations between Burt’s structural constraint and hub node measures

(e.g., betweenness centrality, closeness centrality, and eigenvector centrality). The correlation coefficients were modest (between -0.636 and -0.188), which implicates that a structural brokerage position and a hub node are conceptually different.

Main Comment #4

The ambivalence score is bimodal (figure S2). Does that create a problem for the Authors' CPM?

Answer to Main Comment #4

We demonstrated that the ambivalence score's bimodal distribution is not a problem because CPM uses the "permutation test." Bishara and Hittner (2012) suggested that the permutation test should be used to test the significance of the correlation between bimodally distributed variables (<https://doi.org/10.1037/a0028087>). Many argue that normality assumptions do not have to be met to draw valid inferences from a permutation test. The probability from a permutation test is computed by comparing the obtained test statistic against the "permutation," rather than the theoretical distribution of the test statistic based on the normality assumption (Blair & Karniski, 1993; Cervone, 1985; Wampold and Worsham, 1986).

After revision (pp. 9, line 223): ... To account for the nonindependence of the leave-one-out rounds, permutation tests were conducted. It should be noted that the permutation test also minimizes the potential statistical inference problem caused by the bimodal distribution of the ambivalence score since the test is computed by comparing the obtained test statistic against the "permutation," rather than the theoretical distribution of the test statistic based on the normality assumption (49). ...

Main Comment 5:

The abstract implicates both mentalizing and moral judgment networks in predicting social network data. However, the introduction and rationale for the study only discusses mentalizing. The disconnect is a little odd. Related, the treatment of moral correlates in the discussion section is rather thin and post-hoc. I would prefer if the Authors remove it. Instead, a focus on mentalizing and related social cognition processes is probably most appropriate in the discussion section.

Answer to Main Comment #5

Your points are well taken, and we removed the part suggesting the role of moral judgment networks in the Abstract and Discussion section. Instead, we focused on the role of mentalizing and social cognition processes as you suggested.

Before revision (Abstract): ... Brain regions that contributed most to the prediction included (1) mentalizing regions known to be recruited in reading and understanding others' belief states; and (2) regions associated with moral judgments or information propagation. ...

After revision (Abstract): ... Brain regions that contributed most to the prediction included mentalizing regions known to be recruited in reading and understanding others' belief states. ...

Before revision: Second, dissimilar attitudes "between the others" imposes a cognitive challenge known as "bridging responsibilities" (46). To cope with bridging responsibilities, extra cognitive capacity in addition to mentalizing capacity would be needed as shown by key predictive regions aside from mentalizing regions. Imagine a triadic relationship wherein person A connects with both person B and person C, who have dissimilar attitudes with each other. Merely understanding the attitudes of person B and person C would not

be enough for person A to maintain ties with both people. Person A would need to be able to “switch” between different cognitive frameworks underlying different attitudes (46, 90, 91) so that he or she can naturally communicate with both person B and person C (72). To resolve conflicts between different attitudes and effectively switch between two attitudes, brain regions implicated in resolving cognitive conflict when provided with conflicting social cues, and moral judgments more broadly, such as dlPFC, TPJ, middle frontal gyrus, and superior frontal gyrus, might be required (73-75). Moreover, person A might need to transmit high volumes of contradicting information and ideas, which stem from heterogeneous attitudes between person B and person C (46, 92); dlPFC and TPJ may be supportive of such information propagation (51, 76-78)..

After revision (pp. 14, line 350): Second, dissimilar attitudes “between the others” imposes a cognitive challenge known as “bridging responsibilities” (32). Imagine a triadic relationship wherein person A connects with both person B and person C, who have dissimilar attitudes with each other. Merely understanding the attitudes of person B and person C would not be enough for person A to maintain ties with both people. Person A would need to be able to “switch” between different cognitive frameworks underlying different attitudes (32) so that he or she can naturally communicate with both person B and person C (52). To resolve conflicts between different attitudes and effectively switch between two attitudes, mentalizing brain regions, such as precuneus, and regions implicated in resolving cognitive conflict when provided with conflicting social cues, such as dlPFC, TPJ, middle frontal gyrus, and superior frontal gyrus, might be required (53-55, 71). For example, Chiang et al., 2020 found that precuneus, dlPFC, middle frontal gyrus, and superior frontal gyrus were recruited when one interacts with two people who are affiliated in different social groups and who have negative relationship (71). Moreover, person A might need to transmit high volumes of contradicting information and ideas, which stem from heterogeneous attitudes between person B and person C (32, 72); dlPFC and TPJ may be supportive of such information propagation (56-59). Our control analyses show that attitudinal diversity score is well-predicted even after participant's own gender role attitude is controlled for. This implies that the identified brain connectivity could be related to a cognitive challenge imposed by dissimilar attitudes “between the others” (triadic challenge) above and beyond dissimilar attitudes between “the self and the other” (dyadic challenge).

Minor comments:

Minor Comment 1:

Is figure S1a mislabeled? Figure S1a is suggested to implicate an individual with high brokerage. However, the ways brokerage is commonly conceptualized is a node that connects two different subgraphs or small worlds. Even more simply, a brokerage node is a hub node. In this paper, the Authors describe attitudinal brokerage, which is an individual that connects two otherwise disconnected subgraphs, which is appropriate. But that is not what is reflected in figure S1a. Seems to me that S1a describes network closure, whereas S1b describes a hub node.

Answer to Minor Comment #1

As you pointed out, figure S1a was mislabeled. We appreciate your careful reading. Person A (Figure S1a) occupies an archetypal closure position, and person B (Figure S1b) is in a structural brokerage position (we changed the term “brokerage position” to “structural brokerage.” Please see our answer to Main Comment #3). We corrected the label.

Minor Comment 2:

When drawing edges between nodes, did both individuals have to indicate a connection in the survey data? Only one individual?

Answer to Minor Comment #2

To draw edges between nodes, at least one individual has to indicate a relationship. In other words, we constructed an undirected social network where the tie between two people is assumed if at least one person nominates another person as a discussion partner.

After revision (pp. 5, line 115): For each discussion partner, participant provided a real name, age, gender, address of residence, and communication frequency (days per year). We constructed an undirected social network where the tie between two people is assumed if at least one person nominates another person as a discussion partner. It is because our survey data measured very strong social ties: “spouse and top-five important discussion partners”. Therefore, we assumed that both “someone who is indicated by me as important discussion partner” and “someone who indicates myself as an important discussion partner” are important social network members.

Minor Comment 3:

What is the reliability estimate of the two items in the gender roles instrument? Given that it is just two items, reporting a Spearman-Brown coefficient is probably best (see: <https://doi.org/10.1007/s00038-012-0416-3>)

Answer to Minor Comment #3

We added the Spearman-Brown coefficient in the manuscript (Spearman-Brown coefficient = 0.222). Although two items have been the frequently used measures and were found to be reliable (Davis, Greenstein, 2009; Kangas, Rostgaard, 2007), the Spear-Brown coefficient was not so high in our study. We suspect that two items capture two different components of gender role attitudes in modern rural settings in South Korea.

After revision (pp. 5, line 129): ... The gender role attitude score was the average of the recoded responses to the first and second questions. The two items have been the most frequently used gender role attitude measures (44). The reliability estimate of the two items was also high in the previous studies (45). However, in our study, the reliability estimate of the two items was not so high (Spearman-Brown coefficient=0.222). We suspect that the first and the second questions may capture different dimensions of gender egalitarianism in the context of modern Korean rural villages. Nevertheless, we used the composite score because we believe both dimensions are essential components of gender egalitarianism. ...

Minor Comment 4:

If gender role attitudes is the average of responses to two 5-point likert scales, then the midpoint would be 2.5. Why is it that, in equation one (ambivalence), egalitarian social network members are defined as those who scored 3, and not simply above the midpoint, on the gender role attitudes measure?

Answer to Minor Comment #4

Response categories to our 5-point Likert scales are 1=Strongly agree, 2=Agree, 3=Neither agree nor disagree, 4=Disagree, 5=Strongly disagree, so the midpoint is 3. In the manuscript, we clarified the range of responses.

After revision (pp. 6, line 149): Note that response categories to gender role attitudes score are 1=Strongly agree, 2=Agree, 3=Neither agree nor disagree, 4=Disagree, 5=Strongly disagree and thus, the midpoint is 3.

Minor Comment 5:

On p.9, line 228, the Authors write: "To complement this theory-driven approach..." Saying "theory driven" seems a bit strong here. I'm willing to believe that the Authors conduct a hypothesis driven analysis. But I see nothing in the paper's rationale that implies a strong theory (see main comment #1).

Answer to Minor Comment #5

We agree with your concern and thus, we used the term "hypothesis-driven approach" rather than "theory-driven approach".

After revision (pp. 10, line 257): To complement this hypothesis-driven approach that used mentalizing brain connectivity defined using a meta-analytical approach as a predictor of attitudinal diversity of social network members,

Minor Comment 6:

I see that the Authors make their raw data available. Is there any reason the code is also not available?

Answer to Minor Comment #6

We uploaded the code on GitHub to replicate our analyses.

After revision (Data accessibility): Data, codebook, and the entire study codebook of Korean Social Life, Health, and Aging Project are available at https://osf.io/azrsy/?view_only=d6dd38ffbd7244e6b0cf9928cdeefe3c. Also, the code to replicate our analyses is available at <https://github.com/JunsolKim/neural-and-social-correlates-of-attitudinal-brokerage>.

Referee: 2:

Comments to the Author(s)

This paper reports on an impressive dataset in which the team recorded attitudinal information (about gender attitudes) from nearly all of the elders in two Korean villages and scanned the brains of a subset of these participants with fMRI. The dataset is unique and the questions the authors have asked are important. The results are intriguing. That said, several additional clarifications and analyses would strengthen the paper.

Conceptual questions:

Conceptual question 1:

*Prior work suggests that brain activity and connectivity in the mentalizing system are associated with social network position. In this dataset, it isn't clear whether that is the case. Is connectivity in the mentalizing system related to their measure of brokerage or other network position variables? This seems directly relevant to the interpretation of the brain-attitudinal diversity results, as well as the brain*brokerage interaction.*

Answer to Conceptual question 1

Thank you for pointing out the possibility that brain functional connectivity and social network position

are correlated. Additional analyses revealed that connectivity in the mentalizing system was not significantly correlated to other network position variables. As shown in Table S2, mentalizing brain connectivity was not correlated with network size, structural constraint, betweenness centrality, closeness centrality, eigenvector centrality at the $p < 0.05$ level. We added this result to the manuscript.

Furthermore, in control analyses (Table S3), we additionally controlled for social network position variables such as structural constraint, betweenness centrality, closeness centrality, and eigenvector centrality. However, the association between the mentalizing brain connectivity and attitudinal diversity scores (ambivalence score, standard deviation) remained significant even after controlling social network position variables.

After revision (Supplementary Information: pp. 8, line 194): Previous works suggested that brain activity and connectivity in the mentalizing system is associated with social network position such as structural brokerage (19). Therefore, we examined whether mentalizing brain connectivity was directly correlated to Burt's structural constraint or other social network position variables (betweenness centrality, closeness centrality, eigenvector centrality). However, mentalizing brain connectivity was correlated with none of these variables at $p < 0.05$ level (Table S6). Also, the association between the mentalizing brain connectivity and attitudinal diversity scores (ambivalence score, standard deviation) remained significant even after controlling social network position variables.

After revision (pp. 10, line 241): As shown in Table S3, mentalizing brain connectivity significantly and positively predicted two types of attitudinal diversity scores even after controlling for confounding variables such as socio-demographic factors (age, sex, years of education), social network characteristics (social network size, average communication frequency, structural constraints, betweenness centrality, close centrality, eigenvector centrality), health (Mini-Mental State Exam [MMSE], subjective health), personality (agreeableness, extraversion, neuroticism, openness to experience, conscientiousness), participant's own gender role attitudes, head motion (maximum frame-wise displacement and mean frame-wise displacement), and township of residence. Following previous studies, we controlled for each confounding variable one-by-one while controlling for age and sex by default (48, 50).

Conceptual question 2:

It also seems important to know whether the participant's own baseline attitudes relate to their network's attitudinal diversity. For example, someone with a moderate attitude (in the middle of the scale) might have more ability to connect with different positions, and this might also moderate the mentalizing \diamond ambivalence/SD prediction, or someone with more extreme positions might look like they have more diversity because they are bridging moderate and extreme views. Can you clarify whether a participant's own attitudes are related to any of the key predictors or outcomes and whether the main results hold when controlling for the participant's own attitudes?

Answer to Conceptual question 2

Your question requested two separate answers.

(1)

Can you clarify whether a participant's own attitudes are related to any of the key predictors or outcomes?

As you suggested, we added the test to confirm whether the participant's own baseline attitudes relate to their network's attitudinal diversity and yes, there exist the association. Thanks for your precious suggestion. We regressed attitudinal diversity scores on the participant's own gender role attitude and the results are shown in Table S3 and Figure S3. The result revealed that traditional attitudes were correlated with attitudinal diversity scores in our sample.

As shown in Table S3 and Figure S3, we tested the possibility that the relationship between participant's own baseline attitudes and network's attitudinal diversity is u-shaped (e.g., Is the participant's own modest (middle) attitude or extreme attitude related to network's attitudinal diversity?). However, as shown in Figure S3, the relationship between participant's baseline attitudes and network's attitudinal diversity was shown to be linear.

After revision (pp. 7, line 164): ... Table S1 presents the descriptive statistics of the participants. Also, as shown in Table S2 and Fig. S3, traditional people were more likely to connect with people having diverse attitudes. We believe, as shown in Fig. S4, the plausible explanation would be that there were more egalitarian people than traditional people in our sample.

(2)

whether the main results hold when controlling for the participant's own attitudes?

In control analyses (Table S4), we controlled for participant's own gender role attitudes. The prediction performance of mentalizing brain connectivity was (marginally) significant even after controlling for participant's own gender role attitudes. Thanks to your suggestion, our result could be strengthened and more robust. We appreciate it.

Before revision: As shown in Table S2, mentalizing brain connectivity significantly and positively predicted two types of attitudinal diversity scores even after controlling for confounding variables such as socio-demographic factors (age, sex, years of education), social network characteristics (social network size, average communication frequency, structural constraints), health (Mini-Mental State Exam [MMSE], subjective health), personality (agreeableness, extraversion, neuroticism, openness to experience, conscientiousness), and township of residence. Following previous studies, we controlled for each confounding variable one-by-one while controlling for age and sex by default (64, 65).

After revision (pp. 10, line 241): As shown in Table S3, mentalizing brain connectivity significantly and positively predicted two types of attitudinal diversity scores even after controlling for confounding variables such as socio-demographic factors (age, sex, years of education), social network characteristics (social network size, average communication frequency, structural constraints, betweenness centrality, close centrality, eigenvector centrality), health (Mini-Mental State Exam [MMSE], subjective health), personality (agreeableness, extraversion, neuroticism, openness to experience, conscientiousness), participant's own gender role attitudes, head motion (maximum frame-wise displacement and mean frame-wise displacement), and township of residence. Following previous studies, we controlled for each confounding variable one-by-one while controlling for age and sex by default (48, 50).

Conceptual question 3:

Related to the point above, in the discussion, you note that there are two kinds of computations to resolve (differences between my own and others' attitudes and then differences between friends attitudes) but this isn't really tested empirically. It seems like the data you have would allow you to separate this, for example by controlling for participants' own attitudes and showing that the relationships go above and beyond their own initial position, or by directly focusing on whether the same neural processes are associated with distance from the participants' own attitudes to those of their connections vs. the diversity of the network per se. The authors may also have other ways of addressing this point, but I thought their discussion of it was relevant and that this could benefit from empirical investigation.

According to your suggestion, we added an empirical investigation to show that there are two kinds of computations to resolve. Specifically, we additionally controlled for participants' own attitudes in our control analyses. Our control analyses show that attitudinal diversity scores are well-predicted even

after the participants' own gender role attitude is controlled for. This confirms the existence of a cognitive challenge imposed by dissimilar attitudes “between the others” above and beyond dissimilar attitudes between “the self and the other”. Again, we appreciate for your valuable suggestion.

Before revision: Second, dissimilar attitudes “between the others” imposes a cognitive challenge known as “bridging responsibilities” (46). To cope with bridging responsibilities, extra cognitive capacity in addition to mentalizing capacity would be needed as shown by key predictive regions aside from mentalizing regions. Imagine a triadic relationship wherein person A connects with both person B and person C, who have dissimilar attitudes with each other. Merely understanding the attitudes of person B and person C would not be enough for person A to maintain ties with both people. Person A would need to be able to “switch” between different cognitive frameworks underlying different attitudes (46, 90, 91) so that he or she can naturally communicate with both person B and person C (72). To resolve conflicts between different attitudes and effectively switch between two attitudes, brain regions implicated in resolving cognitive conflict when provided with conflicting social cues, and moral judgments more broadly, such as dlPFC, TPJ, middle frontal gyrus, and superior frontal gyrus, might be required (73-75). Moreover, person A might need to transmit high volumes of contradicting information and ideas, which stem from heterogeneous attitudes between person B and person C (46, 92); dlPFC and TPJ may be supportive of such information propagation (51, 76-78)..

After revision (pp. 14, line 350): Second, dissimilar attitudes “between the others” imposes a cognitive challenge known as “bridging responsibilities” (32). Imagine a triadic relationship wherein person A connects with both person B and person C, who have dissimilar attitudes with each other. Merely understanding the attitudes of person B and person C would not be enough for person A to maintain ties with both people. Person A would need to be able to “switch” between different cognitive frameworks underlying different attitudes (32) so that he or she can naturally communicate with both person B and person C (52). To resolve conflicts between different attitudes and effectively switch between two attitudes, mentalizing brain regions, such as precuneus, and regions implicated in resolving cognitive conflict when provided with conflicting social cues, such as dlPFC, TPJ, middle frontal gyrus, and superior frontal gyrus, might be required (53-55, 71). For example, Chiang et al., 2020 found that precuneus, dlPFC, middle frontal gyrus, and superior frontal gyrus were recruited when one interacts with two people who are affiliated in different social groups and who have negative relationship (71). Moreover, person A might need to transmit high volumes of contradicting information and ideas, which stem from heterogeneous attitudes between person B and person C (32, 72); dlPFC and TPJ may be supportive of such information propagation (56-59). Our control analyses show that attitudinal diversity score is well-predicted even after participant's own gender role attitude is controlled for. This implies that the identified brain connectivity could be related to a cognitive challenge imposed by dissimilar attitudes “between the others” (triadic challenge) above and beyond dissimilar attitudes between “the self and the other” (dyadic challenge).

Conceptual question 4:

*Greater clarification of the theoretical argument linking the brain activity to the network variables would help. For example, the authors argue in support of their observed interaction between mentalizing connectivity and brokerage “he or she still needs to be surrounded by people with different attitudes to exercise that neural capacity” – but an alternative argument would be that people who have certain neural tendencies might seek out different views, or that people who are surrounded by people with different views might develop different mentalizing tendencies. I think it would be helpful to understand whether different model specifications with these same variables give any insight into whether they are all equally plausible or whether the specific order proposed by the authors is most likely given the data (i.e., if you test a model where mentalizing is predicted by an interaction between attitudinal diversity and brokerage, is the fit worse than when mentalizing is the predictor? What about if you look at the interaction between mentalizing and attitudinal variables on brokerage? Etc.). This might also help clarify the argument. Related to this in some places the authors seem to frame the logic like a mediation (network-brain-attitudes) but then test moderation of attitudes ~ brain*network position.*

As you suggested, we tested two alternative models. First, we examined whether the interaction of social network position and attitudinal diversity predicts mentalizing (i.e., mentalizing ~ network position*attitudinal diversity). Second, we investigated whether the interaction of mentalizing and attitudinal diversity predicts social network position (i.e., network position ~ mentalizing*attitudinal diversity). It turns out that our original model specification that the interaction of mentalizing and social network position predicts attitudinal diversity fits better than two alternative models.

After revision: To show that the results were consistent even when using continuous variables, we examined the moderating effects of Burt's structural constraint (continuous) on the association between mentalizing brain connectivity (continuous) and attitudinal diversity of social network members (continuous). As shown in Table S7 and Fig. S11, occupation of a structural brokerage position significantly moderated the association between mentalizing brain functional connectivity and ambivalence scores ($p=0.057$). Also, a structural brokerage position significantly moderated the association between mentalizing brain functional connectivity and standard deviation ($p=0.023$). Consistent with the results in the Main Manuscript – Social Correlates section, participants who were simultaneously in the high brain functional connectivity group (higher mentalizing brain connectivity) and the structural brokerage group (lower structural constraint) showed high ambivalence scores and standard deviations.

It should be noted that social network position was correlated with mentalizing brain connectivity in the previous literature (19). Therefore, In Table S7, we also tested two alternative model specifications. In the first alternative model, mentalizing was predicted by the interaction between attitudinal diversity and structural brokerage position (Burt's structural constraint). The table confirms that the statistical fit measured by r-square was worse than the original model specification. Second, the model that predicted structural brokerage based on the interaction between mentalizing brain connectivity and attitudinal diversity was also tested. Similarly, the statistical fit was worse than the original model specification.

Conceptual question 5:

I appreciated the authors use of open science practices such as posting their data on OSF. I apologize if I missed it, but I didn't see a protocol document/ full study code book or place to see what else was collected and to get more detail about the full measures that produced the data. Can the authors post such a list (of all the measures collected)? Since there is no pre-registration it is difficult to tell whether the attitudinal measure might be the only proxy they have for attitudinal diversity or whether other related variables should be checked. Specifically, the authors argue that resting connectivity within the mentalizing system is associated with attitudinal diversity of participants' social networks, but this is inferred from only one kind of attitude (i.e., attitudes about gender). Do you have any other types of attitudes that were measured where you can see if these effects are robust for different kinds of attitudes? If not, the discussion should acknowledge this limitation. Please also add in summary info in data that are available on OSF to make reproducibility easier (e.g., a study code book). It would also be helpful to include analysis code as well so that it is clear how the data produce the outcomes reported.

Answer to Conceptual question 5

Thank you for your comments regarding data sharing. As you recommended, we shared the codebook of our variables and the entire study codebook. We also uploaded the code on GitHub to replicate our analyses shown in the manuscript.

After revision (Data Accessibility): Data, codebook, and the entire study codebook of Korean Social Life, Health, and Aging Project are available at https://osf.io/azrsy/?view_only=d6dd38ffbd7244e6b0cf9928cdeefe3c. Also, the code to replicate our analyses is available at <https://github.com/JunsolKim/neural-and-social-correlates-of-attitudinal-brokerage>.

Furthermore, in the Discussion section, we mentioned the limitation that gender role attitudes were the only attitudinal measures in our dataset (Korean Social Life, Health, and Aging Project dataset). We are looking forward to a future study equipped with diverse attitude measures.

After revision (pp. 16, line 406): Third, we measured attitudinal diversity by using only one attitudinal measure, gender role attitudes, because this was the only attitudinal measure in our dataset. Future studies may use other sets of attitudinal or belief measures to examine whether our results can be generalized (See the full codebook available at https://osf.io/azrsy/?view_only=d6dd38ffbd7244e6b0cf9928cdeefe3c.)

Conceptual question 6:

If the whole brain connectivity is a significant predictor of the outcome, what does that mean/ can we conclude anything about mentalizing specifically? What if you take random subsets of nodes that are the same size as the mentalizing network? Is mentalizing above what you'd get by chance? Does mentalizing connectivity predict above and beyond the whole brain connectivity?

Answer to Conceptual question 6

We agree with your point and thus, added an analysis to show that mentalizing connectivity predicts above and beyond the whole-brain connectivity. To do so, we used the analytical strategy suggested by Feng et al., 2018. We estimated the prediction performance of “lesioned connectivity”. Lesioned connectivity refers to whole-brain connectivity excluding edges in mentalizing brain connectivity. The result shows that the prediction performance of lesioned connectivity is significantly lower than the prediction performance of mentalizing brain connectivity regarding ambivalence score but not with standard deviation. In that case, we can conclude that mentalizing brain connectivity plays a particularly important role in predicting outcome at least regarding ambivalence score.

In Figure S8, the edges in mentalizing brain connectivity were colored red. The edges in the lesioned connectivity were colored yellow. The prediction performance of the lesioned connectivity regarding ambivalence score was substantially lower than mentalizing brain connectivity. We added this result in the Supplementary Information - Prediction using whole-brain connectivity section.

Before revision: To complement this theory-driven approach that used mentalizing brain connectivity defined using a meta-analytical approach as a predictor of attitudinal diversity of social network members, we used whole-brain resting-state functional connectivity as a predictor. As a result, even without using pre-defined mentalizing brain connectivity, whole-brain resting-state functional connectivity positively predicted ambivalence scores and standard deviations (see Supplementary Information text).

After revision (pp. 10, line 257): To complement this hypothesis-driven approach that used mentalizing brain connectivity defined using a meta-analytical approach as a predictor of attitudinal diversity of social network members, we used whole-brain resting-state functional connectivity as a predictor. As a result, we found that the prediction performance of mentalizing brain connectivity was above and beyond whole-brain resting-state functional connectivity (see Supplementary Information text-8. Prediction using whole-brain connectivity).

After revision (Supplementary Information: pp. 9, line 218): To show that the prediction performance of mentalizing connectivity is above and beyond whole brain connectivity, we used the “lesioned connectivity” approach. The lesioned connectivity refers to whole-brain connectivity excluding edges in mentalizing brain connectivity (20). If the prediction performance of the lesioned connectivity is lower than the prediction performance of mentalizing brain connectivity, we can conclude that mentalizing brain connectivity plays a particularly important role in predicting attitudinal diversity scores. As a result, when predicting the ambivalence score, the prediction performance of the lesioned connectivity ($r = 0.1102$, $p = 0.256$; $MAE =$

0.8846, $p = 0.201$) was substantially lower than the prediction performance of mentalizing brain connectivity ($r = 0.2301$, $p = 0.046$; MAE = 0.8305, $p = 0.030$). However, when predicting the standard deviation, prediction performance of lesioned connectivity ($r = 0.1967$, $p = 0.074$; MAE = 0.8188, $p = 0.126$) was similar to the prediction performance of mentalizing brain connectivity ($r = 0.2033$, $p = 0.061$; MAE = 0.8190, $p = 0.114$). We can conclude that mentalizing brain connectivity plays a particularly important role in predicting outcome at least regarding ambivalence score.

Methods clarifications:

Methods clarifications 1:

The authors note that they defined the ROI using neurosynth from studies that frequently used the term “mentalizing,” but don’t specify which type of map on neurosynth. Please provide more detail about the ROI definition.

Answer to Methods clarifications #1

We admit that the original description was not satisfactory. We used the “association test map” of the term “mentalizing.” In the association test map, mentalizing-related brain regions were identified by integrating previous studies about mentalizing. Specifically, the association test map was created based on the two-way ANOVA testing for the presence of a non-zero association between “mentalizing” term use and voxel activation.

After revision (Supplementary Information: pp. 4, line 79): We identified 51 mentalizing-related brain regions among the 227 regions from Shen’s whole-brain parcellation atlas. To do so, we used a mentalizing mask, which is the association test map of the term “mentalizing” from the Neurosynth meta-analytic tool (<http://neurosynth.org/analyses/terms/mentalizing/>) In the association test map, mentalizing-related brain regions were identified by integrating previous studies about mentalizing. Specifically, the association test map was created based on the two-way ANOVA testing for the presence of a non-zero association between “mentalizing” term use and voxel activation in the previous studies. The mentalizing mask included 46,943 voxels in the vicinity of 6,824 voxels consistently activated in 151 studies that frequently used the term “mentalizing,” applying a rigorous FDR statistical threshold of $q < 0.01$ (7). After downloading the mentalizing mask, we identified 51 “mentalizing” regions out of 227 regions in Shen’s whole-brain parcellation atlas that contain more than 20 voxels in the “mentalizing” mask. All 51 regions are shown in Fig. S5. Given these 51 “mentalizing” brain regions, we identified a mentalizing brain network consisting of 10,251 edges between the 51 “mentalizing” regions and all 227 whole-brain regions, in line with previous studies (8).

Methods clarifications 2:

How was the sub-population of scanned participants selected? In the supplemental materials you note some exclusion criteria, but was everyone who didn’t have cognitive impairment or the criteria you screened for scanned?

Answer to Methods clarifications #2

To select sub-population, 316 people out of the original population ($n=1508$) were selected via quota sampling based on age, gender, subjective health, and social network size. Then we excluded people based on exclusion criteria delineated in supplemental materials (e.g., cognitive impairment).

Before revision: ... Initially, resting-state fMRI (functional magnetic resonance imaging) data ($n=194$; Village K: $n=72$, Village L: $n=122$) were acquired from a subpopulation from the Korean Social Life, Health and Aging Project (KSHAP). To select participants for resting-state fMRI, a line of screening tests was conducted. ...

After revision (Supplementary Information: pp. 2, line 21): ... Initially, resting-state fMRI (functional

magnetic resonance imaging) data (n=194; Village K: n=72, Village L: n=122) were acquired from a subpopulation from the Korean Social Life, Health and Aging Project (KSHAP). To select participants for resting-state fMRI, we first selected 316 individuals via quota sampling based on age, gender, subjective health, and social network size. Then a line of screening tests was conducted. . . .

Methods clarifications 3:

Social network size was measured as the number of people connected to each participant in the village-wise complete social networks. Is that based on who they nominated, who nominated them, both, something else?

Answer to Methods clarifications #3

We included both who they nominated and who nominated them. It is because our survey data measured very strong social ties: “spouse and top-five important discussion partners”. Therefore, we assumed that both “someone who is indicated by me as important discussion partner” and “someone who indicates myself as an important discussion partner” are important social network members. We added this demonstration in the manuscript.

After revision (pp. 5, line 115): We constructed an undirected social network where the tie between two people is assumed if at least one person nominates another person as a discussion partner. It is because our survey data measured very strong social ties: “spouse and top-five important discussion partners”. Therefore, we assumed that both “someone who is indicated by me as important discussion partner” and “someone who indicates myself as an important discussion partner” are important social network members.

Minor:

Minor 1:

The text “including six degrees of separation between people in the United States” was a little confusing to me. It makes it seem like there are 6 degrees of separation between the topics that follows in the list too. Consider re-phrasing.

Answer to Minor 1

As you recommended, we re-phrased “six degrees of separation between people in the United States” to “acquaintance networks between people in the United States of America.”

After revision (pp. 2, line 31): . . . This “small-worldness” characteristic has been observed in numerous kinds of networks: social, technological, physical, and biological networks (3), including acquaintance networks between people in the United States of America (4), scientific collaboration (5), the internet (6, 7), the power grid (1), airline traffic (8), the structure and conformation space of polymers (9, 10), metabolic pathways (11), and brain networks (12).

Minor 2:

The authors argue that their work is relevant to “preventing political polarization” ∠ I might frame as “reducing” since this work seems unlikely to be able to fully prevent polarization.

When introducing ambivalence, please give intuition about what higher and lower values would mean, what the range is, etc.

Answer to Minor 2

As you recommended, we used the term “reducing” instead of “preventing.” We also added the meaning of higher and lower ambivalence scores and the range of ambivalence scores.

After revision: ... Attitudinal brokerage across such clusters can bridge diverse attitudes within short chains of acquaintances, reducing political polarization and promoting social integration (16).

We also added what higher and lower values of ambivalence score mean and what is the range.

After revision (pp. 6, line 150): Therefore, higher ambivalence score would indicate more diverse attitudes of social network members (i.e., connecting with both traditional and egalitarian people). Given that the maximum number of friends are 10 (See Table S1), the maximum ambivalence score is 1825 $(=(365*5+365*5)/2-|365*5-365*5|)$ and the minimum score is -1825 $(=(365*10+0)/2-|365*10-0|)$.

Minor 3:

What is negative prediction and positive prediction? Do you mean edges in the connectivity graph that are positively connected vs. negatively connected? For example, where you write “On the other hand, neither ambivalence scores ($r = 0.1136$, $p = 0.265$; $MAE = 0.8971$, $p = 0.274$) nor standard deviations ($r = -0.0182$, $p = 0.557$; $MAE = 0.9232$, $p = 0.712$) were negatively predicted by mentalizing brain connectivity.” how does this analysis differ from the one in the prior sentence about positive prediction? Why not include both types of edges in the same model to predict outcomes/ why separate them?

Answer to Minor 3

In CPM (connectome-based predictive modeling) protocol, it is assumed that some edges in the brain connectivity graph can contribute to the positive outcome (i.e., positive network). Simultaneously, some edges can contribute to the negative outcome (i.e., negative network) (Shen et al., 2017). Therefore, the prediction performance is estimated for the respective network. Thus, in our paper, CPM first identifies a set of edges in mentalizing brain connectivity of which strength contributes to attitudinal diversity (high-diversity network). Then CPM identifies a set of edges in mentalizing brain connectivity of which strength contributes to attitudinal homogeneity (low-diversity network). After that, CPM estimates the predictive performance of the high-diversity network and low-diversity network, respectively. High predictive performance of high-diversity network shows that CPM successfully identified edges in mentalizing brain connectivity, which contribute to attitudinal diversity (i.e., diversity was positively predicted by mentalizing brain connectivity). On the other hand, the low predictive performance of low-diversity network shows that CPM “failed” to identify edges in mentalizing brain connectivity contributing to low diversity (i.e., diversity was not negatively predicted by mentalizing brain connectivity).

In short, we could find edges in mentalizing brain connectivity that significantly contribute to attitudinal diversity, which is consistent with our hypothesis. However, we could not find edges in mentalizing brain connectivity that significantly contribute to attitudinal homogeneity. In the Supplementary Information Text – The meaning of positive and negative prediction section, we clarified the definition of positive prediction and negative prediction.

After revision (pp. 7, line 145): In CPM (connectome-based predictive modeling) protocol, it is assumed that some edges in the brain connectivity graph can contribute to the positive outcome (i.e., positive network). Simultaneously, some edges can contribute to the negative outcome (i.e., negative network) (9). Therefore, the prediction performance is estimated for the respective network. Thus, in our paper, CPM first identifies a set of edges in mentalizing brain connectivity of which strength contributes to attitudinal diversity (high-

diversity network). Then CPM identifies a set of edges in mentalizing brain connectivity of which strength contributes to attitudinal homogeneity (low-diversity network). After that, CPM estimates the predictive performance of the high-diversity network and low-diversity network, respectively. The high-diversity network's high predictive performance shows that CPM successfully identified edges in mentalizing brain connectivity, which contribute to attitudinal diversity (i.e., diversity was positively predicted by mentalizing brain connectivity). On the other hand, the low-diversity network's low predictive performance shows that CPM "failed" to identify edges in mentalizing brain connectivity contributing to low diversity (i.e., diversity was not negatively predicted by mentalizing brain connectivity). In short, we could find edges in mentalizing brain connectivity that significantly contribute to attitudinal diversity, which is consistent with our hypothesis. However, we could not find edges in mentalizing brain connectivity that significantly contribute to attitudinal homogeneity.

Minor 4:

For the analysis showing that connectivity in mentalizing regions from one village works in the second village, was this using the exact weights defined in one village or just the same edges but updating the weights?

Answer to Minor 4

We used the same edges and the exact weights in both villages.

After revision (Supplementary Information: pp. 10, line 240): ... We identified a set of edges of which connectivity values positively predicted attitudinal diversity scores for more than 90 percent of leave-one-out cross validation (LOOCV) rounds. Note that the identical edges and weights of mentalizing brain connectivity defined in one village are used to predict attitudinal diversity scores in another village. ...

Minor 5:

For the social correlates section, at first I wondered why the authors dichotomize everything and throw away the continuous info? Why not use continuous mentalizing and brokerage scores? Later, the authors report verifying this with a continuous measure but not clear what they found: "Occupation of a brokerage position could moderate the association between mentalizing brain functional connectivity and ambivalence scores (see Supplementary Information text)." – what does this mean? I wasn't sure which part of the supplementary text addressed this directly. Can you add another sentence or two in the main manuscript to clarify?

Answer to Minor 5

We removed the previous text that describes the results of moderating analyses using continuous variables. Instead, in the Supplementary Information text – 10. Moderating Analyses section, we added more straight-forward interpretation of moderating analyses using continuous variables.

Before revision: To show that the results were consistent even when using continuous variables, we examined the moderating effects of Burt's structural constraint (continuous) on the association between mentalizing brain connectivity (continuous) and attitudinal diversity of social network members (continuous). In particular, we estimated the moderating effects of Burt's structural constraint for every connectivity tie within the mentalizing brain network while applying a false discovery rate (FDR) statistical threshold of $q < 0.1$ using the Benjamini–Hochberg procedure. Occupation of a brokerage position could moderate the association between mentalizing brain functional connectivity and ambivalence scores (see Supplementary Information text).

After revision (pp. 12, line 314): To show that the results were consistent even when using continuous

variables, we examined the moderating effects of Burt's structural constraint (continuous) on the association between mentalizing brain connectivity (continuous) and attitudinal diversity of social network members (continuous) (see Supplementary Information text-10. Moderating effects).

Before revision: For each edge out of 10,251 edges of the mentalizing brain network, we regressed attitudinal diversity scores (ambivalence score and standard deviation) against the connectivity value of each edge, structural constraint, and their interaction term controlling for variables of no-interest which were age, sex, social network size, mean communication frequency, MMSE and village (Table S9). Applying statistical threshold of FDR (false discovery rate) $q < 0.1$ using Benjamini-Hochberg procedure (19), social structural constraint was found to moderate the positive association between ambivalence score and connectivity between left dorsolateral prefrontal cortex (dlPFC; $x = -46.1, y = 28.2, z = 26.8$) and right intraparietal sulcus ($x = 41.4, y = -75.3, z = 28.0$) ($p_{FDR} = 0.098$). That is, this connectivity was positively associated with ambivalence scores only among participants with low structural constraint values who occupied brokerage positions. Also, structural constraint moderated the negative association between ambivalence score and connectivity between right amygdala ($x = 31.2, y = 3.7, z = -21.6$) and left anterior cingulate cortex ($x = -7.4, y = -18.2, z = 30.0$) ($p_{FDR} = 0.098$). That is, this connectivity was also negatively associated with ambivalence scores only among participants with low structural constraint values who occupied brokerage positions.

After revision (Supplementary Information: pp. 11, line 248): To show that the results were consistent even when using continuous variables, we examined the moderating effects of Burt's structural constraint (continuous) on the association between mentalizing brain connectivity (continuous) and attitudinal diversity of social network members (continuous). As shown in Table S7 and Fig. S11, occupation of a structural brokerage position significantly moderated the association between mentalizing brain functional connectivity and ambivalence scores ($p = 0.057$). Also, a structural brokerage position significantly moderated the association between mentalizing brain functional connectivity and standard deviation ($p = 0.023$). Consistent with the results in the Main Manuscript – Social Correlates section, participants who were simultaneously in the high brain functional connectivity group (higher mentalizing brain connectivity) and the structural brokerage group (lower structural constraint) showed high ambivalence scores and standard deviations.

Minor 6:

You noted motion parameters were included as nuisance regressors. How many? 6? 12? 24? Did you regress out any physiological noise?

Answer to Minor 6

As shown in our response to Main Comment #2 of the Reviewer # 1, twelve motion parameters (six estimated motion parameters + six first-order temporal derivatives) were controlled for. Also, we regressed out physiological noise as follows. T1-weighted images were segmented into gray matter, white matter, and cerebrospinal fluid (CSF) and warped into MNI standard space. Based on the aCompCor (anatomical component-based noise correction) method, five principal components each from the white-matter and cerebrospinal fluid (CSF) time series were regressed out to exclude non-gray matter BOLD signal. Band-pass temporal filtering (0.008–0.09) was applied to exclude physiological noise.

Before revision: Image preprocessing and denoising was performed using the SPM12 software (Wellcome Department of Imaging Neuroscience, Institute of Neurology, London, UK) with the Conn toolbox 18.a (<http://www.nitrc.org/projects/conn>) default preprocessing pipeline. Functional images were corrected for motion and slice time and warped into MNI standard space. Images were smoothed with an 8-mm full-width half-maximum Gaussian kernel. In addition, the Artifact Detection Tools (https://www.nitrc.org/projects/artifact_detect/) were used to identify motion and signal intensity outlier images. Images with global mean intensity Z-values > 5 and movement > 0.9 mm were identified as outlier

images. Estimated motion parameters and outlier images were used as nuisance covariates in the time-series linear regression. T1-weighted images were segmented into gray matter, white matter, and cerebrospinal fluid (CSF) and warped into MNI standard space. Signals within white matter and the CSF mask were regressed out to exclude non-gray matter BOLD signal. Band-pass temporal filtering (0.008–0.09) was applied to exclude physiological noise.

After revision (Supplementary Information: pp. 3, line 54): Image preprocessing and denoising was performed using the SPM12 software (Wellcome Department of Imaging Neuroscience, Institute of Neurology, London, UK) with the Conn toolbox 18.a (<http://www.nitrc.org/projects/conn>) default preprocessing pipeline. Functional images were corrected for motion and slice time and warped into MNI standard space. Images were smoothed with an 8-mm full-width half-maximum Gaussian kernel. In addition, the Artifact Detection Tools (https://www.nitrc.org/projects/artifact_detect/) were used to identify motion and signal intensity outlier images. Images with global mean intensity Z-values > 5 and movement > 0.9 mm were identified as outlier images. In a denoising process before adjacency matrix construction, six estimated motion parameters, six first-order temporal derivatives and outlier images were used as nuisance covariates in the time-series linear regression. T1-weighted images were segmented into gray matter, white matter, and cerebrospinal fluid (CSF) and warped into MNI standard space. Based on the aCompCor (anatomical component-based noise correction) method, five principal components each from the white-matter and cerebrospinal fluid (CSF) time series were regressed out to exclude non-gray matter BOLD signal (1, 2). Band-pass temporal filtering (0.008–0.09) was applied to exclude physiological noise.

Minor 7:

Sp7, line 149, pval missing a decimal point.

Answer to Minor 7

We corrected the typo as you suggested.

Minor 8:

Fig S1, it looks like the labels are reversed (re A vs. B and brokerage vs. closure)

Please add more to the figure captions to say what you think the take home point is for each graph. For example, Figure S5, what is the red line? Please clarify in the figure legend or label more clearly. Fig. S7. Cluster of similar attitudes – what should the take home point for this graph be? Across all of the figures, more descriptive figure legends would be helpful. Also, please add error bars. Fig S8. What are the red vs. yellow lines in the circle plot.

Answer to Minor 8

As you pointed out, Figure S1 was mislabeled. Person A (Figure S1a) occupies an archetypal closure position, and person B (Figure S1b) is in a structural brokerage position (we changed the term “brokerage position” to “structural brokerage position.” Please see our answer to Referee 1 Main Comment #3). We corrected the label and added error bars across figures. Also, we appended what the red vs. yellow lines in the circle plot are.

Fig. S1. An illustration of a typical brokerage and closure position. Person A occupies an archetypal structural closure position while person B is on a structural brokerage position.

Appendix B

Response to Referees (No. RSPB-2020-2866)

Title: Neural and social correlates of attitudinal brokerage: using the complete social networks of two entire villages

January 4, 2021

Dear Editor,

We deeply appreciate for the opportunity to resubmit our manuscript “Neural and social correlates of attitudinal brokerage: using the complete social networks of two entire villages” (No. RSPB-2020-2866) after the second-round review. The second-round reviews were also very constructive and thus, as described in the point-by-point responses, we have revised the manuscript according to every single suggestion. We believe that the feedbacks have significantly strengthened our manuscript.

Referee: 1

I would like to thank the Authors for their detailed response letter. The additional clarification and analysis resolves nearly all of my concerns. The low reliability value for the gender roles instrument, especially given the critical role of this measure, certainly is a concern that constrains our interpretation of the results. However, and given that this measure has been used in previous research, I do see some value in reusing the same measure in the present study. It is also worth noting that the Authors are transparent about this potential limitation, which ultimately leaves it up to the reader to decide when evaluating this manuscript. I think this is appropriate.

I believe the Authors have produced an impressive manuscript. My concerns are resolved.

Answer to Reviewer 1

We would like to thank Reviewer 1 again for the excellent and constructive feedbacks which substantially strengthened our manuscript.

Referee: 2

Comment #1

The authors did a nice job revising the paper and clarified several of my previous questions. I have a few more suggestions/ clarifications before publication:

The authors used “lesioned connectivity vs mentalizing connectivity” as a way to show that effects are specific to mentalizing. Why not include both in the same model and show that mentalizing explains above and beyond the rest of the brain (i.e.. above and beyond the lesioned connectivity)? Also, when comparing the correlations, there doesn’t seem to be a direct comparison/ test of the difference between the r values. It would be good to test this formally.

Answer to Comment #1

We agreed with your feedback and followed your two suggestions.

First, as you suggested, we included lesioned and mentalizing connectivity in the same regression model to show that mentalizing connectivity explains above and beyond the rest of the brain. In the

regression model that predicts ambivalence scores, the coefficient of mentalizing connectivity was statistically significant but the coefficient of lesioned connectivity was not statistically significant. When predicting standard deviation scores, however, both coefficients of lesioned and mentalizing connectivity were not statistically significant. Based on these results, we could again conclude that mentalizing brain connectivity plays a particularly important role in predicting outcome at least regarding ambivalence score.

Second, to test the difference between predictive accuracies of lesioned and mentalizing connectivity, we tested the difference between the coefficients of lesioned and mentalizing connectivity in the regression models. In the regression model that predicts ambivalence scores, the difference between the coefficients of lesioned and mentalizing connectivity was marginally significant ($p=0.095$). However, regarding standard deviation scores, the difference between the coefficients of lesioned and mentalizing connectivity was not statistically significant ($p=0.936$). It seems that predictive accuracy of the mentalizing connectivity is marginally higher than the one of lesioned connectivity at least regarding ambivalence score.

After revision (Supplementary Information text page 10): To show that the prediction performance of mentalizing connectivity is above and beyond whole brain connectivity, we used the “lesioned connectivity” approach. ... We can conclude that mentalizing brain connectivity plays a particularly important role in predicting outcome at least regarding ambivalence score.

Next, we included “attitudinal diversity scores predicted by lesioned connectivity” and “attitudinal diversity scores predicted by mentalizing connectivity” in the same regression model to predict observed attitudinal diversity scores. As shown in Table S8, when predicting ambivalence score, the coefficient of mentalizing connectivity was statistically significant ($p=0.014$) but the coefficient of lesioned connectivity was not statistically significant ($p=0.593$). And the coefficient of mentalizing connectivity was marginally higher than the coefficient of lesioned connectivity ($p=0.097$). When predicting standard deviation, however, both coefficients of lesioned connectivity ($p=0.275$) and mentalizing connectivity ($p=0.212$) were not statistically significant. And the coefficient of mentalizing connectivity was slightly higher than the coefficient of lesioned connectivity but the difference between the coefficients was not statistically significant ($p=0.929$). Thus, it seemed that the predictive accuracy of mentalizing connectivity was marginally higher than the one of the lesioned connectivity at least regarding ambivalence score.

Comment #2

Related to the question above, several of the tables report both correlation and mean absolute error. You do a nice job explaining to the reader what each represents. Can you add one more sentence to explain to the reader how to interpret differences (e.g., when one is significant and the other isn't)?

Answer to Comment #2

We added how to interpret differences between the correlation coefficient and mean absolute error. In previous studies, the correlation coefficient was widely used to estimate the performance of connectome-based predictive models. However, some argued that a correlation coefficient could be biased and misleading when examining predictive model performance. According to their arguments, a mean absolute error could be a more reliable estimate of predictive model performance when correlation coefficients and mean absolute error produce different results (e.g., when one is significant and the other is not).

After revision (Supplementary Information text page 6): To estimate predictive accuracy, which represents the significance of the association, Pearson's correlation coefficient (r) and mean absolute error (MAE) between the observed attitudinal diversity scores and model-predicted attitudinal diversity scores were calculated. In previous studies, Pearson's correlation coefficient was widely used to estimate the predictive

accuracy of CPM (9, 11-14). However, some argued that Pearson's correlation coefficient could produce biased and misleading estimates of predictive accuracy, suggesting that MAE could be a more reliable measure of predictive accuracy when Pearson's correlation coefficients and MAE produce different results (e.g., when one is significant and the other is not) (15).

Comment #3

The authors write: "As shown in Table S7 and Fig. S11, occupation of a structural brokerage position significantly moderated the association between mentalizing brain functional connectivity and ambivalence scores ($p=0.057$)." But I think significant here means $p<.05$. Consider revising to maybe "marginally moderated" or something like that?

Answer to Comment #3

As you recommended, we replaced the term "significantly" with "marginally".

After revision (Supplementary Information text page 11): As shown in Table S7 and Fig. S11, occupation of a structural brokerage position marginally moderated the association between mentalizing brain functional connectivity and ambivalence scores ($p=0.057$).

Comment #4

Fig S3—The legend says "attitudinal diversity" and y axis says ambivalence. Remind the reader that ambivalence is one operationalization of diversity.

Answer to Comment #4

As you suggested, we clarified in the legend of Fig. S3 that ambivalence is one type of attitudinal diversity scores. We also modified Fig. S3 in a way that clarifies what kind of attitudinal diversity scores is shown in the plot.

Comment #5

The authors used two items re: gender attitudes, and noted: "In our study, the reliability estimate of the two items was not so high (Spearman-Brown coefficient = $=0.222$)."—are the results more strongly driven by one item than the other? If they are capturing different things, it might be useful to have a supplemental analysis that shows how the main results look if you use each item separately, and if one is driving the results, briefly discuss in the discussion.

Answer to Comment #5

Your point was well accepted and thus, we added the supplemental analysis, which shows how the main results look if we use each item of gender role attitudes separately. We examined whether one particular item of our gender role attitude measures mainly drives our results, but this was not the case.

After revision (Supplementary Information text page. 12): Considering that the reliability estimate of our gender role attitude measure was not so high (Spearman-Brown coefficient= 0.222), we examined whether one of the two items in our measure drives our results. As described in the Materials and Methods section, gender role attitude measure consists of two items: "(1) Both the man and woman should contribute to the household income (first item)" and "(2) A man's job is to earn money; a woman's job is to look after the home and family (second item)." Using only one of the two items, we re-calculated attitudinal diversity scores of

social network members. After that, in the regression model, we included both “attitudinal diversity scores calculated using the first item” and “attitudinal diversity scores calculated using the second item” to predict mentalizing connectivity.

The results are shown in Table S9. As shown in the table, both kinds of ambivalence scores were significantly associated with mentalizing connectivity. The difference between the two coefficients was not statistically significant ($p=0.668$). Also, both kinds of standard deviation scores were significantly associated with mentalizing connectivity, and the difference between the two coefficients was not statistically significant ($p=0.388$). Therefore, it seemed that our study results were not driven by one particular item of gender role attitude measure.

We briefly discussed the result in our main manuscript.

After revision (Main manuscript page 16): Third, we measured attitudinal diversity by using only one attitudinal measure, gender role attitudes, because this was the only attitudinal measure in our dataset. Future studies may use other sets of attitudinal or belief measures to examine whether our results can be generalized (See the full codebook available at https://osf.io/azrsy/?view_only=d6dd38ffbd7244e6b0cf9928cdeefe3c). Considering that the reliability estimate of our gender role attitude measure was not so high, we examined whether one of the two items in the measure drove our results. However, we found that this was not the case. Both items were statistically significant and the difference between them was not statistically different (See Supplementary information text 11. Additional analyses for our gender role attitude measure).

Comment #6

The authors deleted “Additional studies examining younger populations and/or urban areas in countries with different cultural and social contexts would be helpful to confirm the generality of our findings. The source code to estimate the model to predict attitudinal diversity of social network members employed in this study, along with the corresponding dataset, is available for the purpose of replication.” – I think this text would be useful to keep.

Answer to Comment #6

The only reason we deleted the text was word-count limitation. Now, as you recommended, we added the sentences in the manuscript.

After revision (Main manuscript pp. 16): There were a few noteworthy limitations in this study. First, as the research was conducted on older adults who resided in the rural villages in South Korea, the results may not be generalized to the other types of populations. Additional studies examining younger populations and/or urban areas in countries with different cultural and social contexts would be helpful to confirm the generality of our findings. The source code to estimate the model to predict attitudinal diversity of social network members employed in this study, along with the corresponding dataset, is available for the purpose of replication (https://osf.io/azrsy/?view_only=d6dd38ffbd7244e6b0cf9928cdeefe3c).

Comment #7

Please add code to OSF page—the data are nicely organized and clear, but I didn’t easily find code to reproduce the analyses/results.

Answer to Reviewer 7

We added the code to reproduce the analyses and results to the OSF page.